# Respiratory immunization using antibiotic-inactivated *Bordetella pertussis* confers T cell-mediated protection against nasal infection in mice

Seyed Davoud Jazayeri, Lisa Borkner , Caroline E. Sutton & Kingston H. G. Mills ✉

The Gram-negative bacterium *Bordetella pertussis* causes whooping cough (pertussis), a severe respiratory disease, especially in young children, which is resurgent despite high vaccine coverage. The current acellular pertussis vaccine prevents severe disease but does not prevent nasal infection with *B. pertussis*. This parenterally delivered vaccine induces potent circulating antibody responses but limited respiratory tissue-resident memory T cells and IgA responses. Here we developed a vaccine approach based on respiratory delivery of antibiotic-inactivated *B. pertussis* (AIBP). Ciprofloxacin-treated *B. pertussis* potently activated antigen-presenting cells to drive T cell responses. AIBP immunization via aerosol or intranasal administration conferred a high level of protection against lung and nasal infection. The AIBP vaccine induced *B. pertussis*-specific interleukin (IL)-17-producing CD4 tissue-resident memory T cells that recruited neutrophils to the respiratory tract. Protection was abrogated by depletion of CD4 T cells or neutralization of IL-17 in mice. Unlike a parenterally delivered whole-cell pertussis vaccine, which induced high levels of serum IL-1β, IL-6, tumour necrosis factor and C-reactive protein, aerosol immunization with the AIBP vaccine did not promote systemic pro-inflammatory responses. We present preclinical evidence of a safe and effective respiratory-delivered vaccine platform for inducing T cell-mediated sterilizing immunity against a respiratory pathogen.

Certain vaccine-preventable infectious diseases are still poorly controlled, in part because parenterally delivered vaccines fail to induce protective immunity at mucosal surfaces, the primary site of infection with many pathogens. However, intranasal delivery of vaccines can induce local immunity in the respiratory tract[1]. Intranasal live attenuated vaccines have been licensed for influenza virus[2] and tested in phase 2 clinical trials against *Bordetella pertussis*[3]. However, attenuated vaccines carry risks associated with infection and colonization with live microorganisms, especially in immunocompromised individuals, and may also modulate immune responses in the nose early in life through alterations to the nasal microbiota[4]. Intranasally delivered protein subunit vaccines require potent mucosal adjuvants, which can be associated with adverse reactions[5].

We have developed an alternative vaccine platform, based on antibiotic-inactivated bacteria, specifically designed for delivery by respiratory routes. Our aim was to apply this platform in the

Immune Regulation Research Group, School of Biochemistry and Immunology, Trinity Biomedical Sciences Institute, Trinity College Dublin, Dublin, Ireland.
✉e-mail: kingston.mills@tcd.ie

design and testing of a new vaccine against *B. pertussis*, where there is an unmet medical need for an improved vaccine[6]. There is also a well-established murine model for *B. pertussis* respiratory infection in which vaccine-induced protection correlates with vaccine efficacy in humans[7]. The Gram-negative bacterium *B. pertussis* infects the upper and lower respiratory tracts causing whooping cough, which is associated with considerable morbidity in children and adults and can be fatal in infants[8]. Whole-cell pertussis (wP) vaccines introduced in the 1940s substantially reduced the incidence of pertussis[9], but were associated with serious systemic reactions, including febrile seizures[10]. Concerns about their side effects led to the development of acellular pertussis (aP) vaccines, based on two to five purified *B. pertussis* antigens, pertussis toxin (PT), filamentous haemagglutinin (FHA), pertactin and fimbrial proteins 2 and 3 (refs. [11,12]). However, since the introduction of the aP vaccines, there has been an increase in the incidence of pertussis during the past decades[13,14], with very recent epidemics in many industrialized countries[15,16].

Although current aP vaccines induce potent serum antibody responses, immunity wanes rapidly after immunization[17]. Furthermore, studies in animal models have shown that aP vaccines do not prevent nasal colonization and transmission of *B. pertussis*[18,19], which is consistent with asymptomatic transmission of *B. pertussis* in fully vaccinated humans[20]. This has been linked with a failure of the aP vaccines to induce local antibodies or T cells in the respiratory tract[18,21]. We have reported that interleukin (IL)-17-producing CD4 tissue-resident memory ($T_{RM}$) cells have a critical role in nasal clearance of *B. pertussis* in a murine model[22]. Moreover, protection of mice against lung infection is mediated by T helper 1 ($T_H$1) and T helper 17 ($T_H$17) cells[23] and antibodies[24]. However, while current parenterally delivered alum-adjuvanted aP vaccines induce potent circulating serum IgG antibodies and $T_H$2-biased responses, they do not induce IgA and are less effective at generating $T_H$1, $T_H$17 or respiratory $T_{RM}$ cells in mice, baboons or humans[18,19,21,25].

Here we have examined the potential of intranasally delivered ciprofloxacin-inactivated bacteria as a mucosal vaccine against *B. pertussis*. We show that immunization of mice by aerosol or intranasal administration of an antibiotic-inactivated *B. pertussis* (AIBP) vaccine induced *B. pertussis*-specific $T_H$1- and $T_H$17-type CD4 $T_{RM}$ cells and IgA in the respiratory tissue and protected mice against lung and nasal infection with *B. pertussis*. Our findings provide evidence that antibiotic-treated bacteria may be an effective vaccine platform for intranasal immunization against respiratory pathogens in humans.

## Results

### Preparation and characterization of the AIBP vaccine
Ciprofloxacin is a broad-spectrum antibiotic that inhibits bacterial DNA synthesis by rapidly inhibiting the activity of DNA gyrase or DNA topoisomerase IV, thus disrupting DNA replication and repair processes[26]. We found that ciprofloxacin at 0.1 mg ml$^{-1}$, 0.25 mg ml$^{-1}$ or 0.5 mg ml$^{-1}$ completely inactivated *B. pertussis* at $6 \times 10^7$ colony forming units (CFU) ml$^{-1}$ after 2–4 h (Fig. 1a). The ciprofloxacin-treated bacteria had intact membrane morphology, and a proportion were elongated but not lysed (Extended Data Fig. 1a). Treatment of *B. pertussis* with levofloxacin, another fluoroquinolone antibiotic, had a similar effect on bacterial morphology (Extended Data Fig. 1b). By contrast, *B. pertussis* treated with the broad-spectrum antibiotic chloramphenicol had lost their membrane structure, consistent with lysis of the bacteria (Extended Data Fig. 1b).

We prepared the AIBP vaccine by treating *B. pertussis* from an overnight liquid culture ($6 \times 10^7$ CFU ml$^{-1}$) with ciprofloxacin at 0.25 mg ml$^{-1}$ for 3 h, followed by 2 washes in 1% casein solution. Inactivation of the bacteria was confirmed by lack of growth on Bordet–Gengou (BG) agar for 3–5 days. Furthermore, no live *B. pertussis* could be detected in the lungs or nose 24 h and 72 h after aerosol

delivery of the AIBP vaccine to mice, showing that AIBP does not colonize the respiratory tract, whereas significant CFU were detected in mice given an aerosol of live *B. pertussis* (Fig. 1b).

We assessed the potential safety of the AIBP vaccine. Aerosol administration of the AIBP vaccine did not significantly increase inflammatory cytokine concentrations in the serum over those detected in mice immunized with phosphate buffered saline (PBS (Fig. 1c). By contrast, immunization of mice with the wP vaccine led to significant increases in the concentrations of IL-1β and tumour necrosis factor (TNF) in serum of mice at 4 h and IL-6 at 4 h and 24 h (Fig. 1c). Furthermore, immunization with the wP vaccine resulted in significantly elevated concentrations of C-reactive protein (CRP) in the serum, whereas serum CRP concentrations were not significantly increased in mice immunized with the AIBP vaccine (Fig. 1d). We also showed that the number of neutrophils in the spleen was not enhanced 24 h or 48 h after immunization with the AIBP vaccine, whereas neutrophils were significantly elevated 24 h and 48 h after immunization with the wP vaccine (Fig. 1e). Furthermore, the number of T cells in the spleen was not enhanced 24 h or 48 h after immunization with the AIBP vaccine (Extended Data Fig. 2a). PT is a major contributor to pertussis disease. We detected high concentrations of PT in the lungs of mice following aerosol administration of live *B. pertussis*, but not following aerosol administration of the AIBP vaccine (Fig. 1f). To address the longer-term safety of the vaccine, we examined body weight changes and showed that mice immunized with the AIBP vaccine gained weight to the same extent as mice immunized with PBS (Extended Data Fig. 2b). These findings show that unlike the wP vaccine, the AIBP vaccine does not have any features associated with potential toxicity in humans.

### The AIBP vaccine activates antigen-presenting cells
Dendritic cells (DCs) have a central role in priming naive T cells. We assessed the effect of the AIBP vaccine on DC maturation. The AIBP vaccine enhanced expression of major histocompatibility complex class II (MHCII), CD40 and CD80 on bone marrow-derived DCs, and this was significantly greater than that induced with the wP vaccine (Fig. 2a). The AIBP vaccine promoted production of the T cell-polarizing cytokines IL-1β, IL-12p70 and IL-23, which was significantly greater than that induced with the wP vaccine (Fig. 2b). We next assessed production of T cell-polarizing cytokines in vivo at the site of immunization. Aerosol immunization with the AIBP vaccine significantly enhanced the concentrations of IL-1β and IL-23 in the lungs 4 h after administration (Fig. 2c), similar to or greater than that induced by parenteral immunization with the wP vaccine (Fig. 2c). Aerosol delivery of the AIBP vaccine, but not the wP vaccine, induced significant IL-6 production in the nasal tissue 4 h after administration (Supplementary Fig. 1). These findings show that aerosol delivery of the AIBP vaccine induces production of $T_H$1- and $T_H$17-polarizing cytokines at the site of immunization in the respiratory tract, but not inflammatory cytokines in the circulation, which can cause side effects.

Finally, we assessed the ability of the AIBP vaccine to stimulate *B. pertussis*-specific T cell responses in vitro. *B. pertussis*-specific CD4 T cells enriched from spleens of convalescent mice, with residual antigen-presenting cells (APCs), secreted IL-17 when cultured with AIBP, and this was significantly greater than observed following culture with the wP vaccine (Fig. 2d). We also compared the AIBP vaccine with heat-killed *B. pertussis* (HKBP) over a range of antigen concentrations. *B. pertussis*-specific CD4 T cells, purified from spleens of convalescent mice, secreted IL-17 and interferon-γ (IFNγ) when cultured with AIBP in the presence of APCs (irradiated spleen cells), and this was significantly greater than that induced with HKBP, especially at low antigen concentrations (Extended Data Fig. 3). Collectively, our findings show that the AIBP vaccine induces DC maturation and production of T cell-polarizing cytokines and activates APC to drive $T_H$1 and $T_H$17 responses.

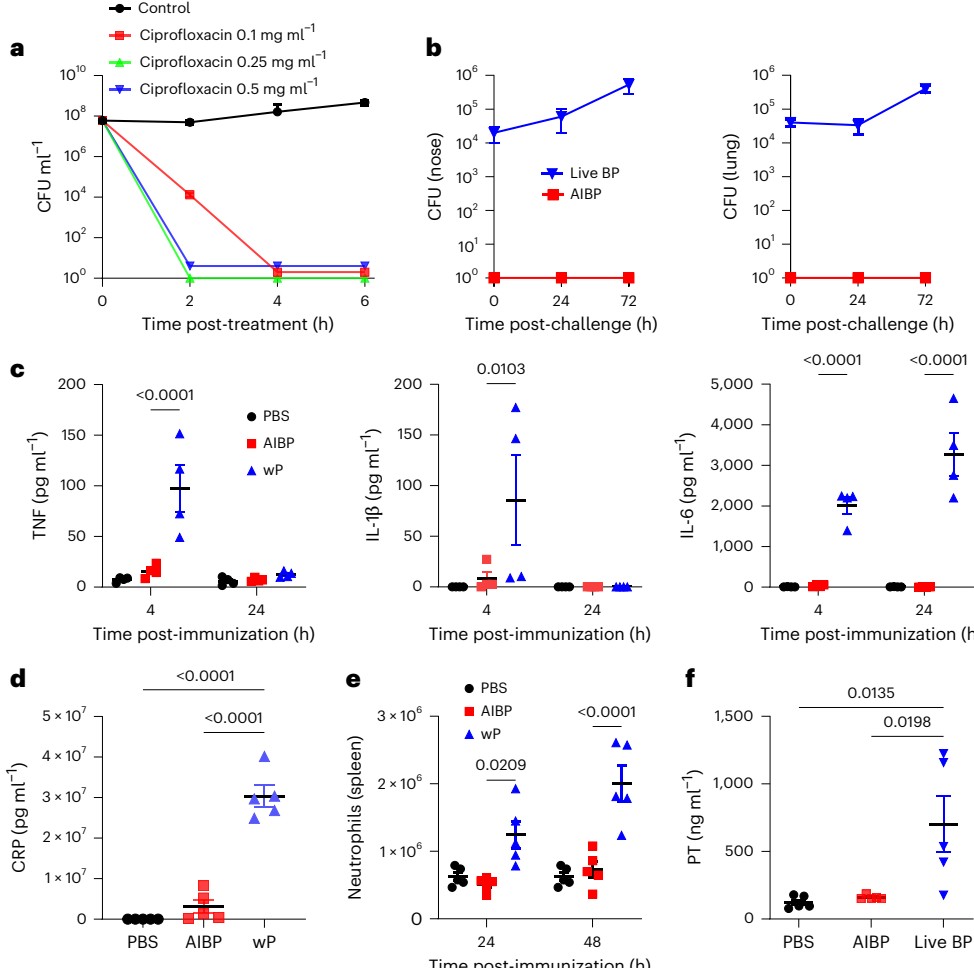

**Fig. 1 | Preparation, characterization and safety of the AIBP vaccine.**
**a**, Survival of *B. pertussis* cultured in S&S medium at a starting concentration of $6 \times 10^7$ CFU ml$^{-1}$ with 0.1–0.5 mg ml$^{-1}$ of ciprofloxacin or medium only; live bacteria were quantified by performing CFU counts on BG agar plates at 2 h, 4 h and 6 h. Data are the mean ± s.d. of biological replicates ($n = 3$). **b**, Female 6–8-week-old C57BL/6 mice were exposed to an aerosol of live *B. pertussis* or AIBP vaccine, and CFU were assessed in lungs and nasal tissue after 2 h, 24 h and 72 h. Data are the mean ± s.d. of biological replicates ($n = 3$ for each time point). **c–e**, Female 6–8-week-old C57BL/6 mice were immunized by aerosol administration of the AIBP vaccine (equivalent to a culture of $1 \times 10^9$ CFU ml$^{-1}$) or were immunized intramuscularly with a wP vaccine (1/50 human dose), and the concentrations of TNF, IL-1β and IL-6 were quantified in serum after 4 h and 24 h ($n = 4$ for each time point) (**c**), the concentration of CRP was quantified in serum after 48 h ($n = 5$) (**d**) and the number of neutrophils in the spleen was quantified by flow cytometry after 24 h and 48 h ($n = 5$ for each time point) (**e**). **f**, Female 6–8-week-old C57BL/6 mice were exposed to an aerosol of live *B. pertussis* or AIBP vaccine, and after 24 h, the concentration of PT was quantified in the lung by ELISA ($n = 5$). Data in **c–f** are presented as the mean ± s.e.m. of biological replicates ($n = 4$ or 5) shown as individual symbols. Data were analysed by two-way ANOVA followed by Tukey's test for multiple comparisons. *P* values are shown above data compared.

## Aerosol AIBP vaccine potently induces respiratory T$_{RM}$ cells and confers sterilizing immunity in the lungs and nasal tract

We have reported that induction of local T cell responses in the respiratory tract is key to protective immunity against *B. pertussis*[22]; therefore, we first examined the ability of AIBP vaccines, prepared with *B. pertussis* treated with ciprofloxacin or levofloxacin, compared with chloramphenicol, to induce respiratory T$_{RM}$ cells. We also compared the immunogenicity of bacteria treated with ciprofloxacin for 3 h or 24 h. Aerosol immunization of mice with *B. pertussis* inactivated with ciprofloxacin (3 h or 24 h) or levofloxacin resulted in expansion of CD4 T cells and CD4 T$_{RM}$ cells in the lungs and nasal tissue of immunized mice (Extended Data Fig. 4a–d). By contrast, the number of CD4 T cells and CD4 T$_{RM}$ cells in mice immunized with chloramphenicol-inactivated *B. pertussis* was at background levels, like that observed in mice immunized with PBS (Extended Data Fig. 4a–d). Furthermore, aerosol immunization of mice with fluoroquinolone antibiotics generated *B. pertussis*-specific CD4 T$_{RM}$ cells in the lungs that secreted IFNγ and IL-17, whereas immunization with chloramphenicol-inactivated *B. pertussis* did not induce *B. pertussis*-specific T$_{RM}$ cells (Extended Data Fig. 4e,f).

The T cell responses were not significantly different in mice immunized with *B. pertussis* treated with ciprofloxacin for 3 h compared with 24 h (Extended Data Fig. 4a–f).

We next examined the immunogenicity and protective efficacy of aerosol-delivered AIBP vaccine, compared with a current licensed parenterally delivered aP vaccine. CD4 T$_{RM}$ cells were detected in the lungs and nasal tissue after a single aerosol immunization with the AIBP vaccine, and this was significantly enhanced following a booster immunization (Fig. 3a). By contrast, respiratory CD4 T$_{RM}$ cells were not enhanced in mice immunized with the aP vaccines. Furthermore, *B. pertussis*-specific IL-17- and IFNγ-producing CD4 T$_{RM}$ cells were induced in lung and nasal tissues of mice immunized with the AIBP vaccine, but not with the aP vaccine (Fig. 3b). These responses were strongest after two immunizations with the AIBP vaccine. Representative flow cytometry plots are shown in Extended Data Fig. 5a–d.

Immunization of mice with two doses of AIBP vaccine induced potent FHA-specific IgA in the nasal tissue, whereas the aP vaccine failed to induce FHA-specific IgA (Fig. 3c). Two doses of AIBP or aP vaccine induced FHA-specific IgG1 and IgG2c in the serum. By contrast,

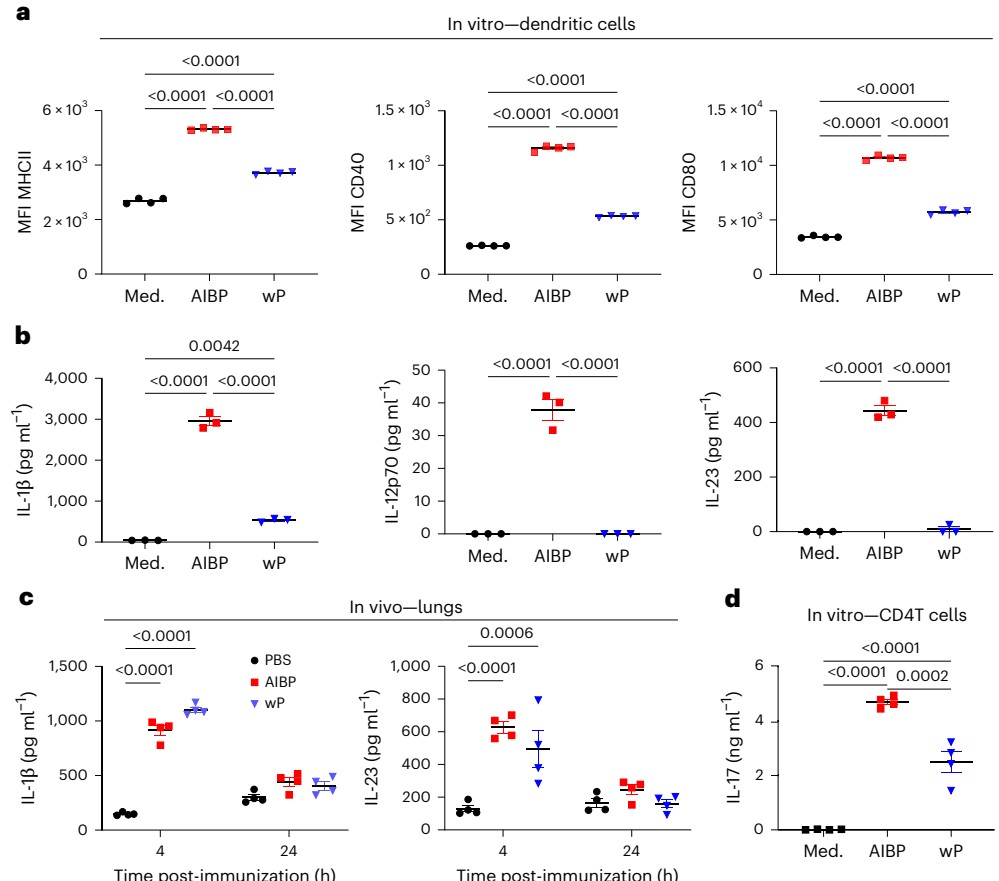

**Fig. 2 | The AIBP vaccine promotes DC maturation and production of $T_H1$- and $T_H17$-polarizing cytokines. a,b,** Bone marrow-derived DCs were stimulated for 24 h with AIBP or wP vaccine (bacterium-to-cell ratio of 10:1) or medium (Med.). **a,** Surface expression of MHCII, CD40 and CD80 was evaluated using flow cytometric analysis gated on live single cells. Results are expressed as mean fluorescence intensity (MFI) and show individual values for four biological replicates. **b,** Concentrations of IL-1β, IL-12p70 and IL-23 in supernatants were quantified by ELISA ($n = 3$). **c,** Groups of eight female 6–8-week-old C57BL/6 mice were immunized by exposure to an aerosol of the AIBP vaccine or intramuscularly with a wP vaccine or PBS. After 4 h and 24 h, concentrations of IL-1β and IL-23 in lung homogenates were quantified by ELISA. **d,** CD4 T cells enriched from the spleen of *B. pertussis* convalescent mice were cultured with AIBP or wP vaccines at concentrations equivalent to 10 bacteria to one DC, and after 3 days, IL-17 was quantified in supernatants by ELISA. Data are presented as the mean ± s.e.m. of four biological replicates shown as individual symbols. Data were analysed by one-way ANOVA followed by Tukey's test for multiple comparisons. *P* values are shown above relevant datasets.

FHA-specific antibodies were undetectable after a single dose of the AIBP vaccine (Fig. 3c).

Mice immunized with 2 doses of the AIBP vaccine completely cleared bacteria from the lungs 14 days after the live *B. pertussis* challenge (Fig. 3d). A single immunization with the AIBP vaccine conferred similar protection against *B. pertussis* infection of the lung to two immunizations with an aP vaccine (Fig. 3d). A single dose of the AIBP vaccine also conferred a high level of protection against infection of the nose, and mice that received two doses had completely cleared the infection from the nose 14 days after the challenge (Fig. 3d). By contrast, immunization with the aP vaccine did not protect against nasal infection with *B. pertussis*.

We also assessed intranasal (i.n.) delivery of the AIBP vaccine by dropping it onto the nose of mice and found that doses equivalent to $3 \times 10^5$ CFU, $3 \times 10^6$ CFU or $3 \times 10^7$ CFU conferred high levels of protection against lung and nasal infection with *B. pertussis*, with the most complete protection achieved with the highest dose (Extended Data Fig. 6).

Collectively, our data show that the AIBP vaccine delivered by the aerosol or i.n. route confers protective immunity against lung and nasal infection, even after a single dose, and this is associated with the induction of potent T cell and antibody responses in the respiratory tract.

Finally, we assessed the durability of immunity induced with the AIBP vaccine. CD4 $T_{RM}$ cells were still elevated in lung and nasal tissues of mice 7 days after the *B. pertussis* challenge of mice immunized 6 months earlier with the AIBP vaccine, and the numbers were significantly greater than in mice immunized with PBS (Extended Data Fig. 7a,b). Furthermore, substantial numbers of *B. pertussis*-specific IL-17- and IFNγ-producing CD4 $T_{RM}$ cells were detectable in lung and nasal tissues. Lower numbers of IL-5-secreting CD4 $T_{RM}$ cells were detected in the lungs, and very low numbers were detected in the nasal tissue (Extended Data Fig. 7c,d). The *B. pertussis* challenge of mice 6 months post-immunization showed a high level of protection against infection of the lungs and nose, with complete bacterial clearance by day 21 (Extended Data Fig. 7e). These findings show that the AIBP vaccine confers durable immunity against *B. pertussis*.

## Efficacy of the AIBP vaccine not affected by priming with an aP vaccine

Existing aP vaccines promote *B. pertussis*-specific $T_H2$, but do not generate strong $T_H1$ or $T_H17$ responses in mice or humans[25,27]. Immunization with the aP vaccine can also suppress the induction of CD4 $T_{RM}$ cells and thereby bacterial clearance after the *B. pertussis* challenge[28]. Here we assessed the induction of T cell subtypes and protective efficacy of two doses of the AIBP vaccine in mice previously immunized with two

doses of an aP vaccine (Fig. 4a). The number of CD4 $T_{RM}$ cells in the respiratory tract was at background levels in mice immunized with the aP vaccine (Fig. 4b). By contrast, there was significant accumulation of CD4 $T_{RM}$ cells in the lungs and nasal tissue of mice immunized with the AIBP, and this was not significantly different in mice immunized with the aP vaccine and boosted with the AIBP vaccine. The aP vaccine failed to generate *B. pertussis*-specific CD4 $T_{RM}$ cells in respiratory tissue (Fig. 4c). By contrast, IL-17- and/or IFNγ-secreting *B. pertussis*-specific CD4 $T_{RM}$ cells were substantially augmented in the lungs and nasal tissue of mice immunized with the AIBP vaccine and these responses were not significantly different in mice immunized with the aP vaccine and boosted with the AIBP vaccine (Fig. 4c).

Immunization with the AIBP vaccine induced nasal IgA, and this response was not affected by previous immunization with the aP vaccine (Fig. 4d). Two doses of the aP vaccine induced *B. pertussis*-specific IgG1 in the serum, and this was not enhanced by boosting with the AIBP vaccine (Fig. 4d). Substantial concentrations of *B. pertussis*-specific IgG2c were induced with two doses of the AIBP vaccine. By contrast, IgG2c responses were close to the background in mice immunized with two doses of the aP vaccine (Fig. 4d). However, IgG2c was induced in mice primed with two doses of the aP vaccine and boosted with the AIBP.

Consistent with the data in Fig. 3, two immunizations with the AIBP vaccine conferred a high level of protection against lung and nasal infection with *B. pertussis* and this was similar to that observed in mice immunized with two doses of the aP vaccine and boosted with two doses of the AIBP vaccine (Fig. 4e). By contrast, immunization with two doses of the aP vaccine alone conferred only modest protection in the lungs, but not in the nose.

We also assessed the effect of previous immunization with two doses of an aP vaccine on the immune responses and protective efficacy of a single dose of the AIBP vaccine. A single dose of the AIBP vaccine induced CD4 $T_{RM}$ cells and IL-17- and/or IFNγ-secreting *B. pertussis*-specific CD4 $T_{RM}$ cells, and this was not significantly different in mice that had previously been immunized with two doses of the aP vaccine (Extended Data Fig. 8a,b). Furthermore, boosting with a single dose of the AIBP vaccine enhanced protection induced with the aP vaccine in the lungs (Extended Data Fig. 8c), whereas the protection against nasal infection induced with a single dose of the AIBP vaccine was similar in mice previously immunized (twice) with the aP vaccine versus PBS (Extended Data Fig. 8c).

Our findings show that previous immunization of mice with two doses of the aP vaccine, which is known to selectively induce $T_H2$ responses[7], does not compromise the ability of one or two doses of the AIBP vaccine to induce CD4 $T_{RM}$ cells and *B. pertussis*-specific IL-17- and IFNγ-secreting CD4 $T_{RM}$ cells or to protect against lung or nasal infection with *B. pertussis*.

**The AIBP vaccine induces stronger $T_{RM}$ responses and confers superior protection against nasal infection than a wP vaccine**
Good-quality wP vaccines are the gold standard for pertussis vaccine efficacy, whereas immunity generated by previous infection is

effective and long lived[22,29,30]. Here we compared the AIBP vaccine with previous infection or immunization with the wP vaccine given by the conventional intramuscular (i.m.) route. CD4 $T_{RM}$ cells accumulated in the lungs and nasal tissue of mice immunized with the AIBP vaccine, and this was significantly stronger than that induced by immunization with the wP vaccine (Extended Data Fig. 9a). IL-17- or IFNγ-secreting *B. pertussis*-specific CD4 $T_{RM}$ cells were significantly higher in the lungs of mice immunized with the AIBP vaccine than in the lungs of mice immunized with the wP vaccine or previously infected (Extended Data Fig. 9b). Furthermore, IFNγ-secreting *B. pertussis*-specific CD4 $T_{RM}$ cells were significantly higher in the nasal tissue of mice immunized with the AIBP vaccine than in the nasal tissue of mice immunized with the wP vaccine (Extended Data Fig. 9b).

Assessment of systemic *B. pertussis*-specific T cell responses revealed that lymph node and spleen cells from mice immunized with two doses of the AIBP vaccine produced substantial quantities of *B. pertussis*-specific IL-17 and IFNγ. The IL-17 production was similar to that induced by previous infection and significantly greater than that induced by two doses of the wP vaccine (Supplementary Fig. 2).

The AIBP vaccine and previous infection induced *B. pertussis*-specific IgA in the nasal mucosa, whereas the wP vaccine failed to induce IgA (Extended Data Fig. 9c). However, i.m. immunization with the wP vaccine generated higher concentrations of *B. pertussis*-specific serum IgG1 and IgG2c than immunization with the aerosol-delivered AIBP vaccine or previous infection (Extended Data Fig. 9c).

The AIBP vaccine, the wP vaccine and previous infection all conferred complete protection against *B. pertussis* infection of the lungs (Extended Data Fig. 9d). However, the best protection against nasal infection was observed with the aerosol-delivered AIBP vaccine. AIBP-immunized mice and previously infected mice had completely cleared the infection from the nose by day 14 and 21, respectively, whereas bacteria were still detectable in the nose 14 days and 21 days after the live *B. pertussis* challenge of mice intramuscularly immunized with the wP vaccine (Extended Data Fig. 9d).

Collectively, our findings show that while the AIBP induced weaker serum IgG responses, it generated more potent IgA, systemic $T_H1$ and $T_H17$ responses, and IL-17- and IFNγ-secreting CD4 $T_{RM}$ cells. Importantly, the aerosol-delivered AIBP vaccine conferred greater protection against infection of the nose than the parenterally delivered wP vaccine or previous infection.

As the respiratory route of immunization may have contributed to the superior efficacy of the AIBP vaccine, we examined the immunogenicity and protective efficacy of the AIBP vaccine compared with those of the wP or aP vaccines, when all vaccines were delivered by the i.n. route. Intranasal immunization with AIBP induced CD4 $T_{RM}$ cells in the lungs and nasal tissue (Fig. 5a). By contrast, respiratory CD4 $T_{RM}$ was at background levels in mice immunized with the wP or aP vaccine by the i.n. route. Furthermore, *B. pertussis*-specific IL-17- and IFNγ-producing CD4 $T_{RM}$ cells were induced in lung and

---

**Fig. 3 | Aerosol-delivered AIBP vaccine induces respiratory $T_H1$ and $T_H17$ $T_{RM}$ cells and confers protection against *B. pertussis* infection of the lungs and nasal cavity.** Female 6–8-week-old C57BL/6 mice were immunized by aerosol administration of the AIBP vaccine (once or twice at 0 week and 4 weeks), an aP vaccine (i.m., twice, 0 week and 4 weeks; 1/50 human dose) or PBS. The mice were aerosol challenged from a culture at $1 \times 10^9$ CFU ml$^{-1}$ of live *B. pertussis* at week 6. On the day of but before the challenge with live *B. pertussis*, mice were injected intravenously with anti-CD45 antibody 10 min before euthanasia (to identify tissue-resident cells), and lung or nasal tissue cells were stained with antibodies specific for $T_{RM}$ cells, or cells were stimulated with HKBP, anti-CD28 and anti-CD49d (both 1 μg ml$^{-1}$) for 16 h, followed by Brefeldin A (5 μg ml$^{-1}$) for the final 4 h of culture before intracellular cytokine staining (ICS) and flow cytometric analysis. **a**, Mean absolute number of CD4 $T_{RM}$ cells (CD45iv$^-$ CD4$^+$CD

44$^+$CD62L$^-$CD69$^+$CD103$^{+/-}$) quantified in lung and nasal tissue by flow cytometric analysis. Data are mean ± s.e.m. ($n = 5$). **b**, Number of *B. pertussis*-specific IFNγ- or IL-17-secreting CD45iv$^-$CD4 $T_{RM}$ cells in lung and nasal tissues determined using ICS and flow cytometry. Data in **a** and **b** are presented as the mean ± s.e.m. of biological replicates shown as individual symbols ($n = 5$). **c**, Before the *B. pertussis* challenge, concentrations of FHA-specific IgA in nasal tissue homogenates and FHA-specific IgG1 and IgG2c in serum were quantified by ELISA. Data are presented as the mean ± s.e.m. of biological replicates ($n = 5$). **d**, Live bacterial loads in lung and nasal tissue were quantified by CFU counts at 2 h and 7 days, 14 days and 21 days after the live *B. pertussis* challenge. Data are presented as the mean ± s.e.m. of biological replicates ($n = 5$). Data were analysed by one-way ANOVA followed by Tukey's test for multiple comparisons. *P* values are shown above relevant datasets.

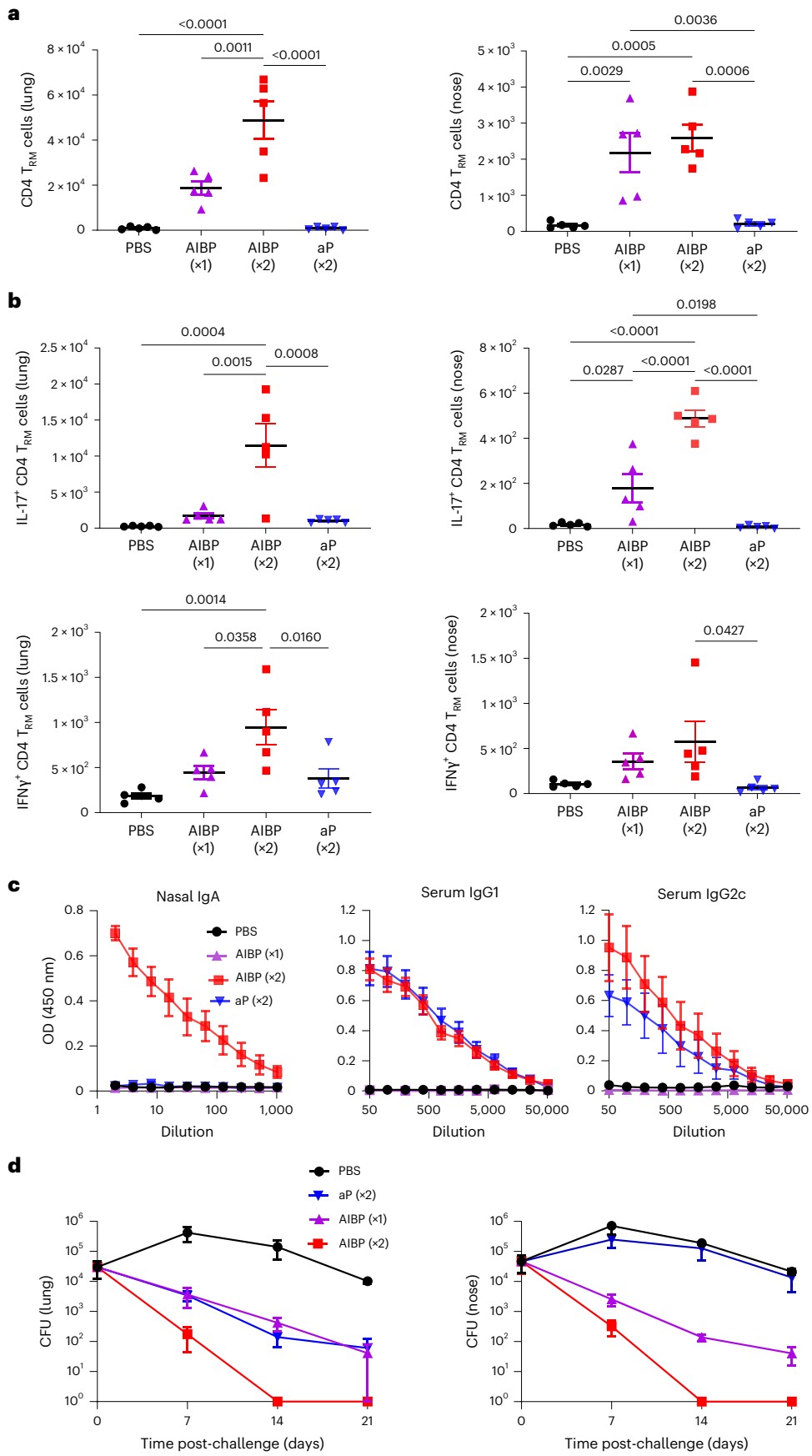

nasal tissues of mice immunized with the AIBP vaccine but were close to background in mice immunized intranasally with the wP or aP vaccine (Fig. 5b).

Immunization of mice with the AIBP vaccine induced potent *B. pertussis*-specific IgA in the nasal tissue and IgG1 and IgG2c in serum. By contrast, antibody responses were weak or undetectable in mice immunized by the i.n. route with the wP or aP vaccine (Fig. 5c).

Intranasal immunizations with the AIBP vaccine conferred a high level of protection against lung and nasal infection with *B. pertussis* (Fig. 5d). By contrast, i.n. immunization with the wP vaccine induced modest protection against lung or nasal infection and this was substantially poorer than that observed with the AIBP vaccine. The aP vaccine failed to protect against lung or nasal infection when administered by the i.n. route (Fig. 5d). These findings show that the superior efficacy of the AIBP vaccine is not solely due to its delivery via the respiratory tract.

## Mechanism of protective immunity induced with the AIBP vaccine

Although antibodies have a role in preventing infection of the lungs in mice[7,31] and maternal antibodies protect against pertussis disease in human infants[32], studies in mice have shown that IL-17-secreting T cells are required for clearance of *B. pertussis* from the nasal tract[22]. Studies in mice, baboons and humans have shown that current a P vaccines selectively induce $T_H 2$ cells and antibody responses, but not respiratory $T_{RM}$ cells, and consequently fail to prevent nasal colonization with *B. pertussis*[7,18,19,25]. We examined the possible role of IL-17-secreting CD4 T cells by depleting CD4 cells or neutralizing IL-17 before and after challenge with *B. pertussis* in mice immunized with one dose of the AIBP vaccine.

Flow cytometry analysis showed that recruitment of CD4 T cells to the lungs and nose in mice immunized with the AIBP vaccine was abrogated in mice treated with the anti-CD4 antibody and was also reduced in mice treated with the anti-IL-17 antibody (Fig. 6a). IL-17 is known to recruit neutrophils, especially Siglec-F⁺ neutrophils, to the respiratory tissue of *B. pertussis*-infected mice. Here we found enhanced recruitment of Siglec-F⁺ neutrophils to the lungs and nasal tissue after the *B. pertussis* challenge of mice immunized with the AIBP vaccine, and this was reversed in mice treated with anti-IL-17 and significantly reduced in mice treated with anti-CD4 (Fig. 6b).

Protection against infection induced with a single dose of the AIBP vaccine was completely abrogated after depletion of CD4 T cells or neutralization of IL-17; the CFU counts in the anti-CD4-depleted mice were similar to those in non-immunized control mice (Fig. 6c).

Our findings show that the AIBP vaccine mediates sterilizing immunity largely via induction of IL-17-secreting CD4 T cells that promote recruitment of Siglec-F⁺ neutrophils to the respiratory tract.

## Discussion

This study describes a new paradigm in vaccination against bacterial infections. In proof-of-principle experiments with ciprofloxacin-treated *B. pertussis*, we show that respiratory delivery of ciprofloxacin-inactivated bacteria is a safe and effective vaccine approach for inducing sterilizing immunity against *B. pertussis* of the

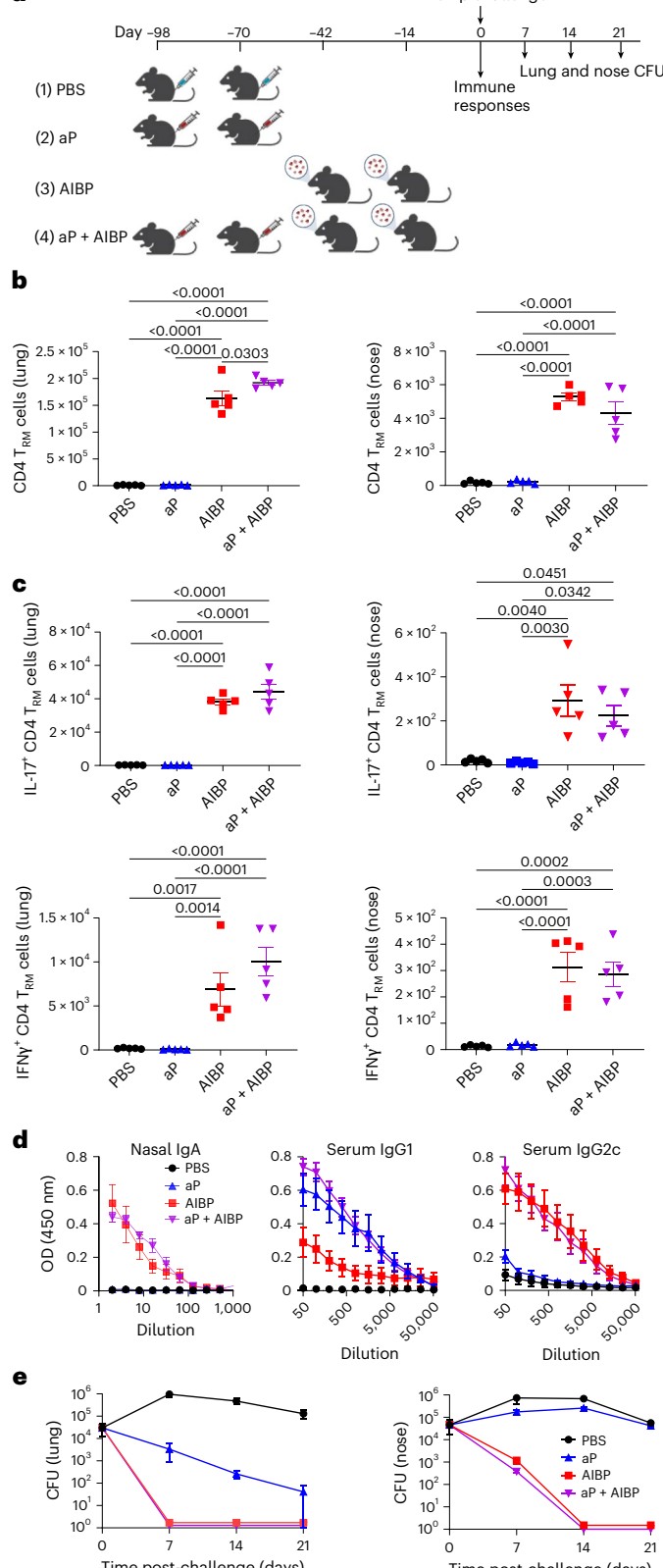

**Fig. 4 | Previous immunization of mice with aP vaccines does not affect the induction of $T_H 1$ and $T_H 17$ cells or the efficacy of the AIBP vaccine. a**, Schematic of the experiment; mice were immunized intramuscularly with the aP vaccine (twice at 0 week and 4 weeks; 1/50 of the human dose) followed by aerosol administration of the AIBP vaccine (twice, 8 weeks and 12 weeks) or given two doses of the aP or AIBP only or PBS. Female 6–8-week-old C57BL/6 mice were aerosol challenged with live *B. pertussis* at week 14. **b,c**, On the day of but before the challenge with live *B. pertussis*, CD4 $T_{RM}$ cells (**b**) and *B. pertussis*-specific IL-17- and IFNγ-producing CD4 $T_{RM}$ cells (**c**) were analysed by ICS and flow cytometry as described in the legend of Fig. 3. Data in **b** and **c** are presented as the mean ± s.e.m. of biological replicates shown as individual symbols (*n* = 5). **d**, On the day before the *B. pertussis* challenge, concentrations of *B. pertussis*-specific IgA, IgG1 and IgG2c in nasal tissue homogenates and serum were quantified by ELISA. Data are presented as the mean ± s.e.m. of biological replicates (*n* = 5). **e**, CFU counts on lung and nasal tissue 2 h, 7 days, 14 days and 21 days post-challenge. Data are presented as the mean ± s.e.m. of biological replicates (*n* = 5). Data were analysed by two-way ANOVA followed by Tukey's test for multiple comparisons. *P* values are shown above relevant datasets. Panel **a** created with BioRender.com.

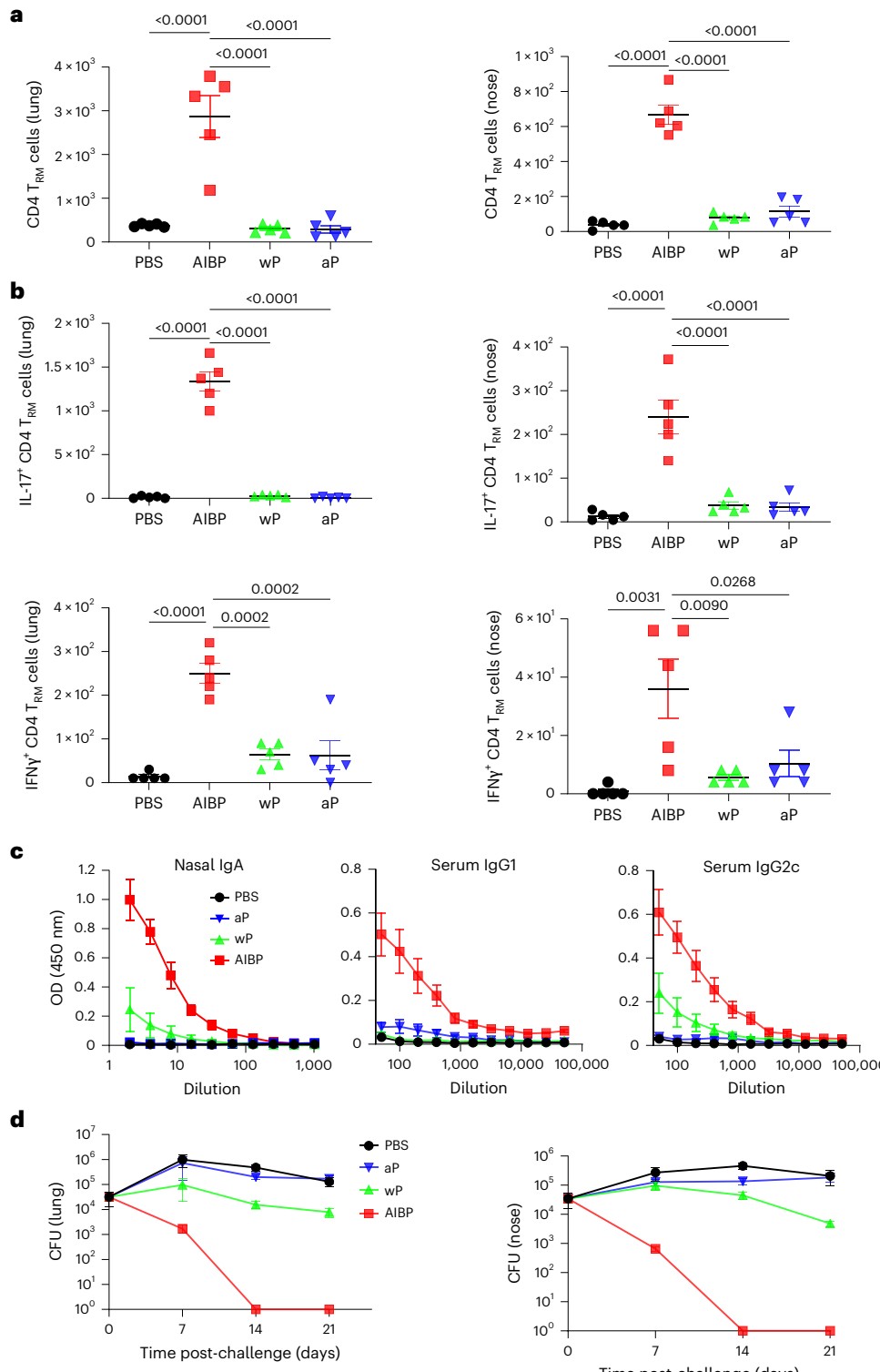

**Fig. 5 | The AIBP vaccine has superior immunogenicity and protective efficacy to wP or aP vaccines delivered by the nasal route.** Female 6–8-week-old C57BL/6 mice were immunized by i.n. administration of AIBP vaccine (equivalent to $1 \times 10^8$ bacteria per mouse), wP vaccine (1/160 human dose, equivalent to $1 \times 10^8$ bacteria per mouse), aP vaccine (1/160 human dose) or PBS twice at 0 week and 4 weeks and were challenged with live *B. pertussis* 2 weeks later. **a**,**b**, On the day of but before the challenge with live *B. pertussis*, CD4 $T_{RM}$ cells (**a**) and *B. pertussis*-specific IL-17- and IFNγ-producing CD4 $T_{RM}$ cells (**b**) were analysed by ICS and flow cytometry as described in the legend of Fig. 3. Data in **a** and **b** are presented as

the mean ± s.e.m. of biological replicates shown as individual symbols (*n* = 5). **c**, On the day before the *B. pertussis* challenge, concentrations of *B. pertussis*-specific IgA in nasal tissue homogenates and IgG1 and IgG2c in serum were quantified by ELISA. Data are presented as the mean ± s.d. of biological replicates (*n* = 5). **d**, CFU counts on lung and nasal tissue 2 h and 7 days, 14 days and 21 days post-challenge. Data are the mean ± s.d. of biological replicates (*n* = 5). Data were analysed by one-way ANOVA followed by Tukey's test for multiple comparisons. *P* values are shown above relevant datasets.

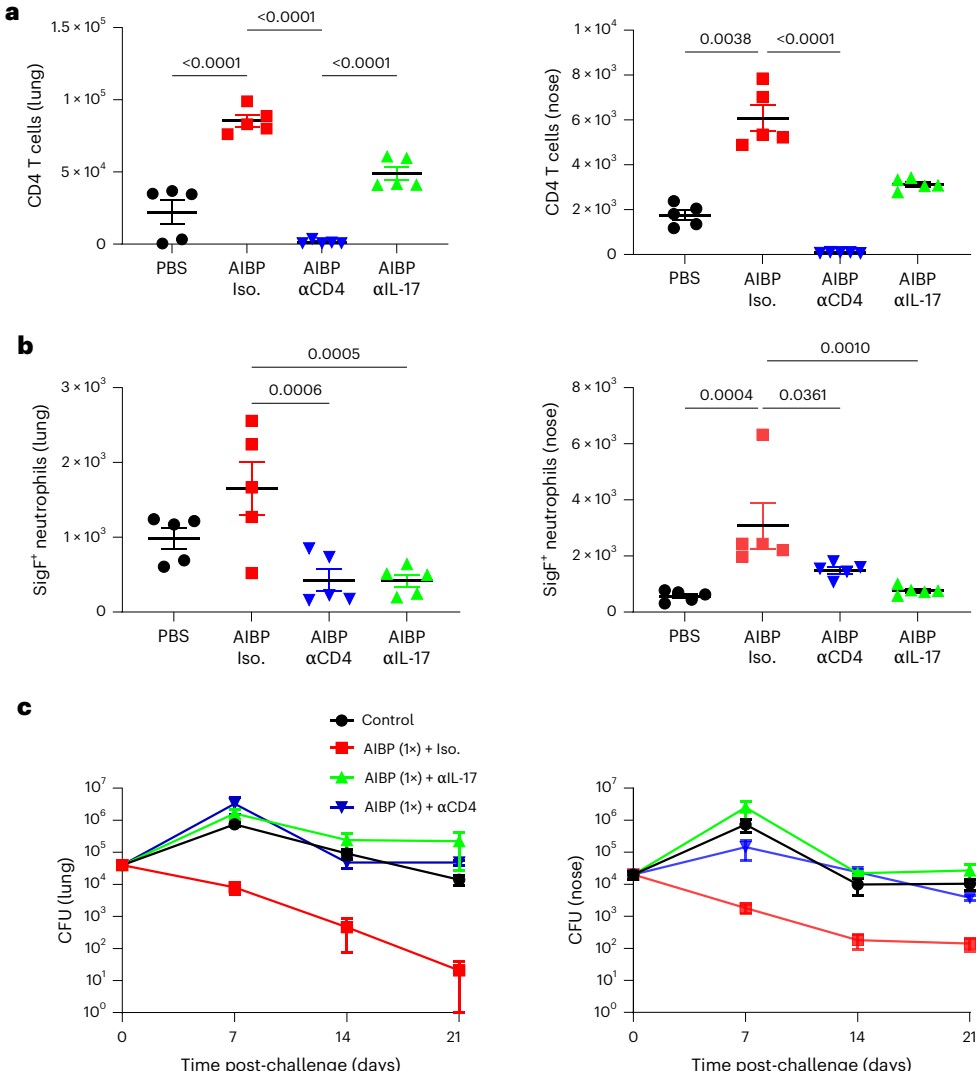

**Fig. 6 | IL-17-secreting CD4 T cells mediate protection induced with the AIBP vaccine.** Female 6–8-week-old C57BL/6 mice were immunized once by aerosol administration of the AIBP vaccine or PBS and challenged by aerosol with live *B. pertussis* at week 6. Groups of mice that had been immunized with the AIBP vaccine were treated with anti-CD4, anti-IL-17 or an isotype (Iso.) control antibody the day before and every 3 days after the challenge with *B. pertussis*. **a**, On the day of but before the challenge with live *B. pertussis*, CD4 cells were quantified in lungs and nasal tissue by flow cytometric analysis. **b**, On the day of but before the *B. pertussis* challenge, Siglec-F⁺ neutrophils were quantified in lung and nasal tissue cells by flow cytometry. Data in **a** and **b** are presented as the mean ± s.e.m. of biological replicates shown as individual symbols (*n* = 5). **c**, CFU counts were performed on lung and nasal tissue 2 h and 7 days, 14 days and 21 days post-challenge. Data are presented as the mean ± s.d. of biological replicates (*n* = 5). Data were analysed by two-way ANOVA followed by Tukey's test for multiple comparisons. *P* values are shown above relevant datasets.

upper and lower respiratory tracts of mice. Delivery of the AIBP vaccine to mice by the aerosol or intranasal route promoted the induction of IgA and CD4 $T_{RM}$ cells in respiratory tissue and conferred protective immunity against *B. pertussis* infection of the lungs and nasal tract, which was mediated largely by IL-17-secreting CD4 T cells and associated recruitment of Siglec-F⁺ neutrophils.

Current aP vaccines are suboptimal, failing to prevent infection of the nasal tract and allowing transmission of *B. pertussis* from immunized individuals[18,19]. Attempts to develop new vaccines against pertussis have focused on live attenuated vaccines[3], outer membrane vesicle vaccines[33] or subunit vaccines administered with adjuvants[34,35]. The attenuated vaccines are the most developed, with BPZE1 showing good efficacy in phase 2 clinical trials[3]. However, the limitations of attenuated vaccines include failures to 'take' owing to pre-existing antibodies[36] and transient colonization of the nasopharynx, which poses potential safety issues in immunocompromised individuals. Our approach of using ciprofloxacin-treated *B. pertussis*

has significant advantages over current aP vaccines and combines the benefits of live attenuated pertussis vaccines and wP vaccines, but with low risks and high immunogenicity when delivered to the respiratory tract. However, the AIBP vaccine has very distinct characteristics and properties from wP vaccines. wP vaccines are prepared by inactivating *B. pertussis* with aldehydes, usually formaldehyde, which cross-links proteins by forming covalent bonds between different amino acid residues. This can adversely affect conformational antibody epitopes and alter protein degradation by endo-lysosomal proteases, thereby affecting antigen processing[37]. Conformational changes to proteins, such as those induced by aldehyde treatment, can adversely affect T cell activation[38]. This may affect the immunogenicity and protective efficacy of the killed bacteria in the wP vaccines. Indeed, the variability in immunogenicity of wP vaccines may reflect the different duration of formaldehyde treatment, which influences the extent of protein antigen cross-linking. By contrast, the bacterial antigens are not modified in the AIBP vaccine, and our data show that

the AIBP vaccine is more effective than the wP vaccine at stimulating APCs and thereby promoting $T_H1$ and $T_H17$ responses. Furthermore, when compared with parenteral or i.n. immunization with the wP vaccine, aerosol or i.n. delivery of the AIBP vaccine induced significantly greater accumulation of $T_H1$- and $T_H17$-type CD4 $T_{RM}$ cells in the respiratory tissue and conferred better protection against *B. pertussis* infection of the nasal cavity.

Ciprofloxacin and other fluoroquinolone antibiotics permeate bacterial membranes and arrest cell division by inhibiting the function of DNA gyrase and/or DNA topoisomerase IV leading to DNA damage[39]. A proportion of the bacteria enlarge after fluoroquinolone treatment, and this may facilitate uptake of the inactivated bacteria by APCs. Furthermore, the AIBP vaccine induced DC maturation and production of the T cell-polarizing cytokines IL-1β, IL-12p70 and IL-23, which was significantly greater than that induced with the wP vaccine. This is consistent with the highly effective priming of IL-17- and IFNγ-secreting respiratory CD4 $T_{RM}$ cells. By contrast, *B. pertussis* treated with the beta-lactam antibiotic chloramphenicol, which induces bacterial cell lysis, failed to induce respiratory $T_{RM}$ cells. Although two doses of the AIBP vaccine prepared by ciprofloxacin treatment were required to induce potent serum IgG and mucosal IgA, a single dose of the AIBP vaccine induced respiratory $T_{RM}$ cells and conferred a high level of protection against *B. pertussis* infection of the lungs and nasal tract. This protection persisted for at least 6 months and was significantly attenuated following depletion of CD4 T cells or neutralization of IL-17. Although we do not rule out a role for mucosal IgA or circulating IgG2c antibodies, which are involved in complement fixation and opsonization, our findings suggest that IL-17-producing CD4 $T_{RM}$ cells are major mediators of protective immunity induced with the AIBP vaccine.

The aerosol-delivered AIBP vaccine has significant advantages over current parenterally delivered aP vaccines. Most importantly, it protects against infection of the nose and lungs and should therefore prevent community transmission of *B. pertussis*, and because it induces *B. pertussis*-specific $T_{RM}$ cells, protective immunity should be sustained. Furthermore, the immunogenicity and protective efficacy of the AIBP vaccine was not affected by previous immunization with two doses of an aP vaccine. The AIBP vaccine was still capable of inducing $T_H1$ and $T_H17$ cells and $T_{RM}$ cells in mice immunized with the $T_H2$-inducing aP vaccine, suggesting that it could be used effectively for a booster vaccine in aP-primed humans.

In terms of safety, aerosol delivery of the AIBP vaccine did not promote systemic production of proinflammatory cytokines or serum CRP, observed with a parenterally delivered wP vaccine, which has been linked to adverse events[40]. Furthermore, it did not colonize in the lungs or nose or secrete PT in immunized mice. It should be possible to further reduce any possible risks associated with the AIBP vaccine by preparing it from *B. pertussis* with reduced lipooligosaccharide content[41] and by deleting or mutating other *B. pertussis* toxins. Another significant advantage of the AIBP vaccine over existing aP vaccines is the ease and cost of production, making it especially attractive for low- and middle-income countries. Respiratory administration of the AIBP vaccine with a simple nebulizer or spray device would allow needle-free delivery, thus promoting vaccine compliance and uptake. Because of the multiple antigens expressed by the AIBP vaccine, it has the capacity to protect against circulating strains with mutations or deletions in *B. pertussis* antigens present in the aP vaccine. Therefore, this technology has considerable potential for the development of more effective vaccines against infection with *B. pertussis* and other respiratory bacteria in humans.

## Methods
### Mice
Our research complies with all relevant ethical regulations. All animal experiments were conducted according to the guidelines and under licences approved by the Health Products Regulatory Authority of Ireland (project licence number AE19136) with previous ethical approval from the Trinity College Dublin Animal Research Ethics Committee. All experiments used female C57BL/6 mice that were 6–10 weeks old at the initiation of the experiment. The mice were purchased from Charles River UK and housed in a specific pathogen-free facility, with a 14-h light–10-h dark cycle, at 20 °C and 40–60% relative humidity, in the Comparative Medicine Unit, Trinity College Dublin.

### Antimicrobial activity of ciprofloxacin against *B. pertussis*
*B. pertussis* 338 from an overnight culture in Stainer–Scholte (S&S) medium was adjusted to a concentration of $6 × 10^7$ colony forming units (CFU) $ml^{-1}$. Freshly prepared ciprofloxacin (Enzo Life Sciences) was diluted in nuclease-free water and added to the bacterial culture at concentrations of 0.1–0.5 mg $ml^{-1}$. The CFU $ml^{-1}$ of live bacteria were quantified by performing CFU counts on BG agar plates after 2 h, 4 h and 6 h.

### Morphology of inactivated *B. pertussis*
*B. pertussis* 338 was inactivated with ciprofloxacin (0.25 mg $ml^{-1}$), levofloxacin (1 mg $ml^{-1}$) or chloramphenicol (100 μg $ml^{-1}$) for 24 h followed by 4% paraformaldehyde fixation (10 min room temperature). The concentration of each antibiotic used was based on the minimum concentration that completely arrested bacterial growth after 3 h. The samples were loaded into a Cytospin-MICROTEKNIK-JP-6 following the manufacturer's instructions and centrifuged at $335 × g$ for 10 min. *B. pertussis* was detected using FM 4-64 dye according to the manufacturer's instructions. Image acquisition was performed using an SP8 confocal microscope (Leica).

### Vaccines and immunization
To prepare the AIBP vaccine, *B. pertussis* 338, from an overnight culture, was adjusted to $6 × 10^7$ CFU $ml^{-1}$ in S&S medium and treated with ciprofloxacin (0.25 mg $ml^{-1}$), levofloxacin (1 mg $ml^{-1}$) or chloramphenicol (100 μg $ml^{-1}$). After 3 h or 24 h of antibiotic treatment (37 °C, 180 rpm), the bacteria were collected by centrifuging at $876 × g$ for 20 min at 4 °C and washed twice with 1% casein solution. The concentration of the collected bacteria was adjusted to $1 × 10^9$ CFU $ml^{-1}$. Inactivation of the bacteria in the AIBP vaccine was confirmed by plating on BG agar and monitoring for any live bacterium after 3–5 days. Mice were immunized once or twice after a 4-week interval by aerosol administration of AIBP using a nebulizer (PARI TurboBOY SX) from a culture at $1 × 10^9$ CFU $ml^{-1}$ over 10 min as described previously[42] or by intranasal administration by placing two 15-μl droplets of the AIBP vaccine ($3 × 10^5$, $3 × 10^6$, $3 × 10^7$ or $1 × 10^8$ bacteria per dose) on the mouse nares. To confirm that there are no live bacteria in the AIBP vaccine, CFU counts were performed on digested nasal tissue or lung homogenates of mice 2 h and 3 days after vaccination. The plates were monitored for any live bacteria after 3–5 days. Alternatively, mice were immunized intramuscularly with a 1/50 human dose of a commercial aP vaccine (Boostrix, GlaxoSmithKline) or wP vaccine (NIBSC code: 8/522). In experiments comparing the efficacy of different vaccines delivered by the i.n. route, we used the AIBP vaccine at $1 × 10^8$ bacteria per dose, the wP vaccine at $1 × 10^8$ bacteria per dose (equivalent to 1/160 human dose) and the aP vaccine at 1/160 human dose. In some experiments, immunized mice were compared with convalescent mice, which were defined as mice that were >60 days after the *B. pertussis* challenge.

### *B. pertussis* respiratory challenge
*B. pertussis* 338 bacteria were grown from frozen stocks for 3 days on BG plates. Bacteria were then collected and cultured in supplemented S&S medium overnight at 37 °C in a shaking incubator at 180 rpm. Bacteria were centrifuged and resuspended in 1% casein solution, and the optical density (OD) was measured at 600 nm. Mice were infected by the aerosol challenge administered using a nebulizer from a culture

at $1 \times 10^9$ CFU ml$^{-1}$ over 10 min as described previously[42]. At intervals after the challenge, whole lung and nasal tissue (nasal cavity and nasal turbinates) were aseptically removed, transferred to a Petri dish and chopped with a scalpel. Lungs and nasal tissues were digested in PBS containing collagenase D (1 mg ml$^{-1}$; Sigma-Aldrich) and DNase I (20 U ml$^{-1}$; Sigma-Aldrich) for 1 h at 37 °C. Cells were centrifuged at $367 \times g$ and supernatants were used to assess CFU counts.

### Inflammatory responses induced by the AIBP vaccine

Mice were immunized either by aerosol administration of the AIBP vaccine (at a concentration equivalent to $1 \times 10^9$ CFU ml$^{-1}$) or i.m. administration of a wP vaccine (1/50 human dose) or were infected with live *B. pertussis* ($1 \times 10^9$ CFU ml$^{-1}$). Mice were euthanized 4 h or 24 h after administration. Serum, lung homogenate and nasal tissue supernatants were collected for analysis of inflammatory cytokines (IL-6, IL-1β and TNF) by enzyme-linked immunosorbent assay (ELISA). CRP in serum was quantified using a CRP ELISA Kit (Proteintech) according to the manufacturer's instructions. PT was quantified in lung homogenates by ELISA as described[43].

### DC stimulation

Murine bone marrow-derived DCs were prepared as described previously[44]. DCs ($1 \times 10^6$) in RPMI 1640 medium (without penicillin and streptomycin) were cultured in 96-well U-bottom plates with the AIBP vaccine or wP vaccine at a concentration equivalent to 10 bacteria to 1 DC for 24 h at 37 °C in a 5% $CO_2$ incubator. MHCII and co-stimulatory molecule expression was evaluated using flow cytometric analysis. Supernatants were collected from cell cultures for quantification of IL-1β, IL-6, IL-12p70 and IL-23 by ELISA.

### Activation of *B. pertussis*-specific $T_H1$ and $T_H17$ cells in vitro

CD4 T cells were enriched from spleens of convalescent mice following *B. pertussis* infection. This cell preparation contained more than 90% CD4 T cells, with residual macrophages and DCs, which can act as APCs. The enriched CD4 T cells were cultured ($0.5 \times 10^6$ cells) with the AIBP or wP vaccines at concentrations equivalent to $2 \times 10^6$ bacteria ml$^{-1}$. After 3 days of culture, the concentrations of IL-17 in supernatants were quantified by ELISA. Alternatively, CD4 T cells were magnetic-activated cell sorting (MACS) purified from spleens of convalescent mice and co-cultured ($1 \times 10^5$ ml$^{-1}$) with increasing concentrations of AIBP or HKBP in the presence of APCs (irradiated spleen cells $2 \times 10^6$ ml$^{-1}$). After 3 days of culture, the concentrations of IL-17 and IFNγ in supernatants were quantified by ELISA.

### In vivo treatment with antibodies

For IL-17 neutralization, mice were injected intraperitoneally with anti-IL-17 antibody (17F3; BioXcell) at 300 µg per mouse. For depletion of CD4 T cells, mice were treated with anti-CD4 antibody (YTS177; BioXcell), 200 µg per mouse intraperitoneally and 100 µg per mouse intranasally (15 µl per nare under anaesthesia) simultaneously 1 day before infection and every 3 days after the *B. pertussis* challenge. A corresponding isotype antibody (Rat IgG2b, κ; BioXcell) was used as a control.

### Quantification of $T_{RM}$ cells

To discriminate circulating from tissue-resident lymphocytes by flow cytometry, mice were injected intravenously with 1.5 µg of PE-conjugated anti-CD45 antibody (30-F11; eBioscience) in 200 µl PBS 10 min before euthanasia. Circulating lymphocytes are labelled CD45i.v.$^+$, while tissue-resident lymphocytes are protected from intravenous (i.v.) labelling and are therefore identified as CD45i.v.$^-$ cells. Lungs and nasal tissues were digested and mashed through a 70-µm strainer, red blood cells were lysed with ammonium chloride–potassium buffer, and cells were centrifuged at $367 \times g$ and used for flow cytometric analysis. Mononuclear cells prepared from lungs or nasal tissue

were incubated with LIVE/DEAD Aqua (1:600; Invitrogen), following by incubating with Fc block (BD Biosciences; 1:50) to block IgG Fc receptors. Surface markers were then stained with fluorochrome-conjugated anti-mouse antibodies specific for CD69 (clone H1.2F3), CD45R/B220 (clone RA3-6B2), CD8 (clone 53-6.7), CD3 (clone 145-2C11), CD4 (clone GK1.5), CD44 (clone IM7), CD62L (clone MEL-14), CD103 (clone M290), Ly6G (clone 1A8), CD11b (clone M1/70), Siglec-F$^+$ (clone 50-1702-82), MHC class II (MHCII; clone M5/114.15.2), CD80 (clone 16-10A1) and CD86 (clone GL-1) from Biolegend, BD Biosciences or Invitrogen and fixed with 2% paraformaldehyde (Thermo Fisher Scientific). Tissue-resident cells were defined through lack of in vivo labelling with anti-CD45. Cells that were CD45i.v.$^-$CD4$^+$CD44$^+$CD62L$^-$ and express CD69, with or without CD103, were considered to be CD4 $T_{RM}$ cells. For detection of cytokine-producing *B. pertussis*-specific T cells, mononuclear cells were stimulated for 16 h with HKBP ($10^5$ CFU ml$^{-1}$), anti-CD28 and anti-CD49d (1 µg ml$^{-1}$; BD Biosciences). Brefeldin A (5 µg ml$^{-1}$) was added for the final 4 h of culture. The cells were fixed, permeabilized and stained using the eBioscience Foxp3/Transcription Factor Staining Buffer Set (Thermo Fisher Scientific) according to the manufacturer's instructions using anti-mouse antibodies specific for IFNγ (clone XMG1.2), IL-5 (clone TRFK5) or IL-17 (clone TC11-18H10). Flow cytometric analysis was performed on an Aurora, and data were acquired using SpectroFlo. Data were analysed using FlowJo software (Tree Star). Flow cytometry gating strategies are shown in Supplementary Fig. 3. Details of all antibodies used are provided in Supplementary Table 1.

### Quantification of antibody responses

Serum samples for analysis of IgG were prepared by centrifugation of clotted whole blood at $5,000 \times g$ at 4 °C for 10 min after bleeding of mice by cardiac puncture. *B. pertussis*-specific IgA was quantified in nasal tissue homogenates, and IgG1 and IgG2c were quantified in serum by ELISA using plate-bound HKBP ($10^7$ CFU ml$^{-1}$) or FHA (1.0 µg ml$^{-1}$) and biotin-conjugated anti-mouse IgA, IgG1 or IgG2c (1:1500 HRP conjugated, Southern Biotech). The reaction was developed using 3,3′,5,5′-tetramethylbenzidine (TMB) and 1 M $H_2SO_4$, and the plates were read using a SpectraMax ABS microplate reader (Molecular Devices) at 450 nm.

### Statistical analysis

Statistical analyses were performed using GraphPad Prism 9.0 Software. Data were presented as mean ± s.e.m. Data were analysed by one-way ANOVA or two-way ANOVA followed by post hoc Tukey's test for multiple comparisons. $P$ values < 0.05 were considered significant.

### Reporting summary

Further information on research design is available in the Nature Portfolio Reporting Summary linked to this article.

## Data availability

Source data are provided with this paper.

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

## Acknowledgements

This study was supported by research grants from Science Foundation Ireland/Research Ireland (16/IA/4468, 22/FFP-A/10297) to K.H.G.M. This publication also emanated from research conducted under the ARC Hub Programme, administered by Research Ireland and co-funded by the Government of Ireland and the European Union through the ERDF Southern, Eastern & Midland Regional Programme 2021–2027 under grant number 23/ARC/11991 at the ARC Hub for Therapeutics.

## Author contributions

Study design: K.H.G.M. and S.D.J. Supervision and funding acquisition: K.H.G.M. Methodology and data collection: S.D.J., C.M.S. and L.B. Writing, review and editing of the paper: K.H.G.M. and S.D.J.

## Competing interests

K.H.G.M. and S.D.J. are inventors on a patent application (PCT/EP2025/065749) on the AIBP vaccine. K.H.G.M. is an inventor on a patent (US Patent Application Number 16/271802) around an adjuvant for a pertussis vaccine and has received research funding and acted as a consultant for vaccine manufacturers. L.B. and C.E.S. do not have any competing interests to declare.

## Additional information

**Extended data** is available for this paper at https://doi.org/10.1038/s41564-025-02166-6.

**Correspondence and requests for materials** should be addressed to Kingston H. G. Mills.

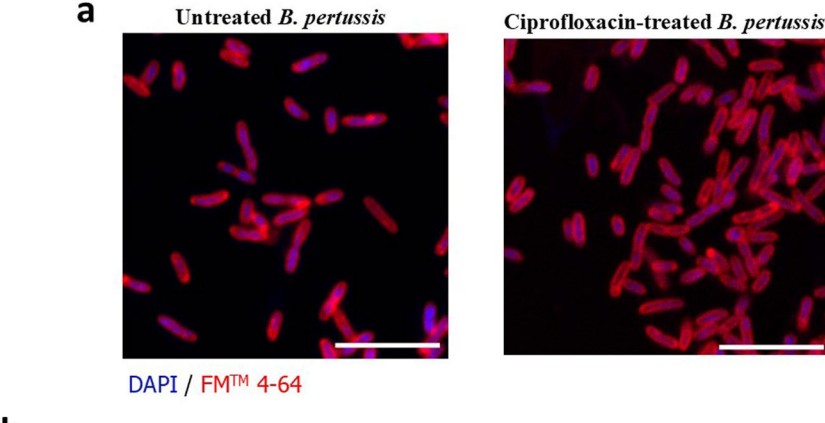

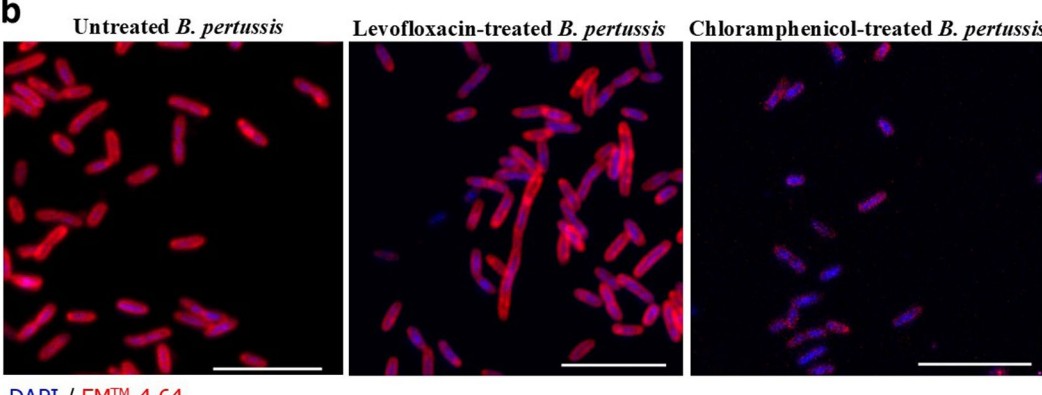

**Extended Data Fig. 1 | Morphology of antibiotic-treated** *B. pertussis.* Representative images of *B. pertussis* culture 24 h after treatment with ciprofloxacin or medium only (untreated) (**a**); this experiment was carried out 3 times with similar results. Representative images of *B. pertussis* culture 24 h after treatment with levofloxacin, chloramphenicol or medium only (untreated) (**b**). This experiment was carried out twice with similar results. Bacteria were fixed with PFA and stained with DAPI (nuclei-blue) and FM™ 4-64 dye (cell membrane of bacteria-red). Scale bar 5 μM.

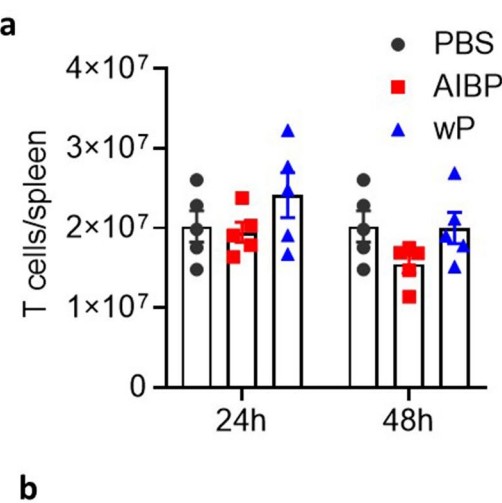

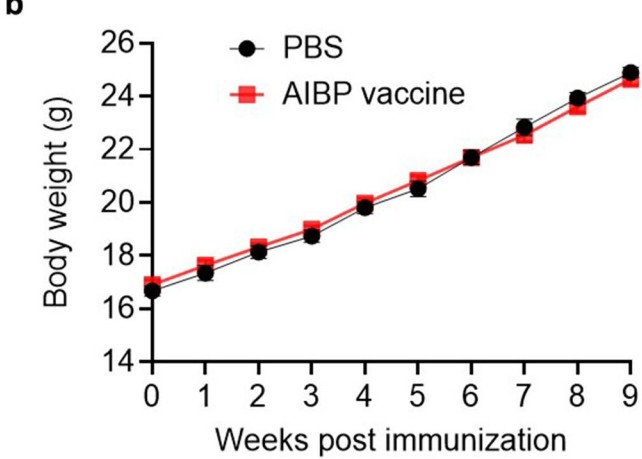

**Extended Data Fig. 2 | Safety testing of the AIBP vaccine. a**, Female 6-8 week old C57BL/6 mice were aerosol immunized with the AIBP vaccine or immunized i.m. with a wP vaccine or with PBS. After 24 and 48 h, the number of T cells in spleen was quantified by flow cytometry. Data are presented as mean ± s.d. for biological replicates shown as individual symbols ($n$ = 5). **b**, Body weight of mice after intranasal immunization (0 and 4 weeks) with the AIBP vaccine or with PBS and *B. pertussis* challenge at week 6. Data are presented as mean ± s.d. for biological replicates ($n$ = 5). Data were analyzed by two-way ANOVAs followed by Tukey's test for multiple comparisons.

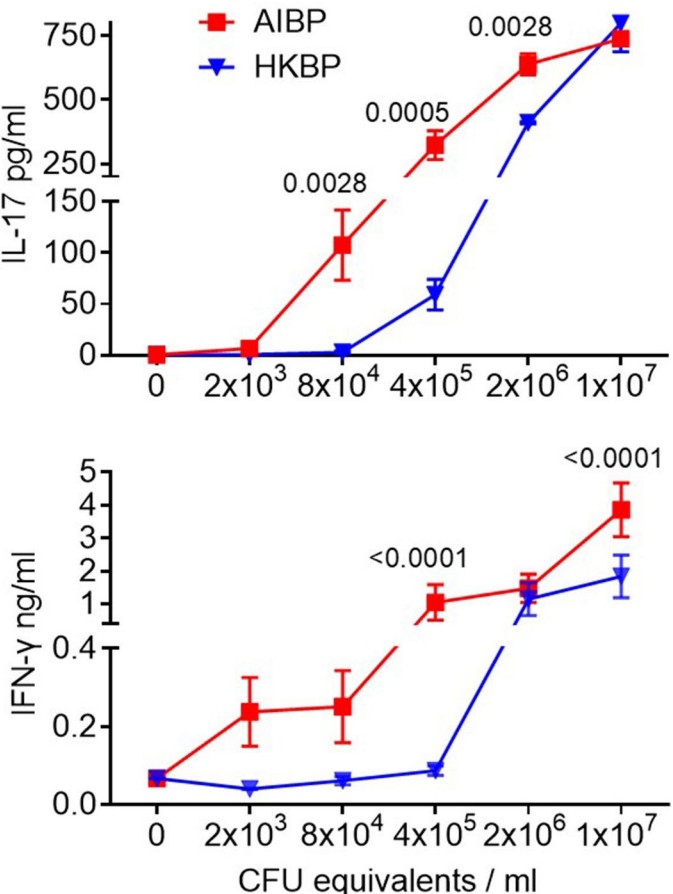

**Extended Data Fig. 3 | AIBP vaccine-activated APC stimulate *B. pertussis*-specific T$_H$1 and T$_H$17 cells *in vitro*.** CD4 T cells purified from the spleen of *B. pertussis* convalescent mice were cultured with APC (irradiated spleen cells, 2 ×10$^6$/ml) and increasing concentrations of AIBP or HKBP. After 3 days, IL-17 and IFN-γ production was quantified in supernatants by ELISA. Data are presented mean ± s.d for biological replicates (*n* = 3). Data were analyzed by two-way ANOVAs followed by Tukey's test for multiple comparisons. *P* values are shown above relevant data sets.

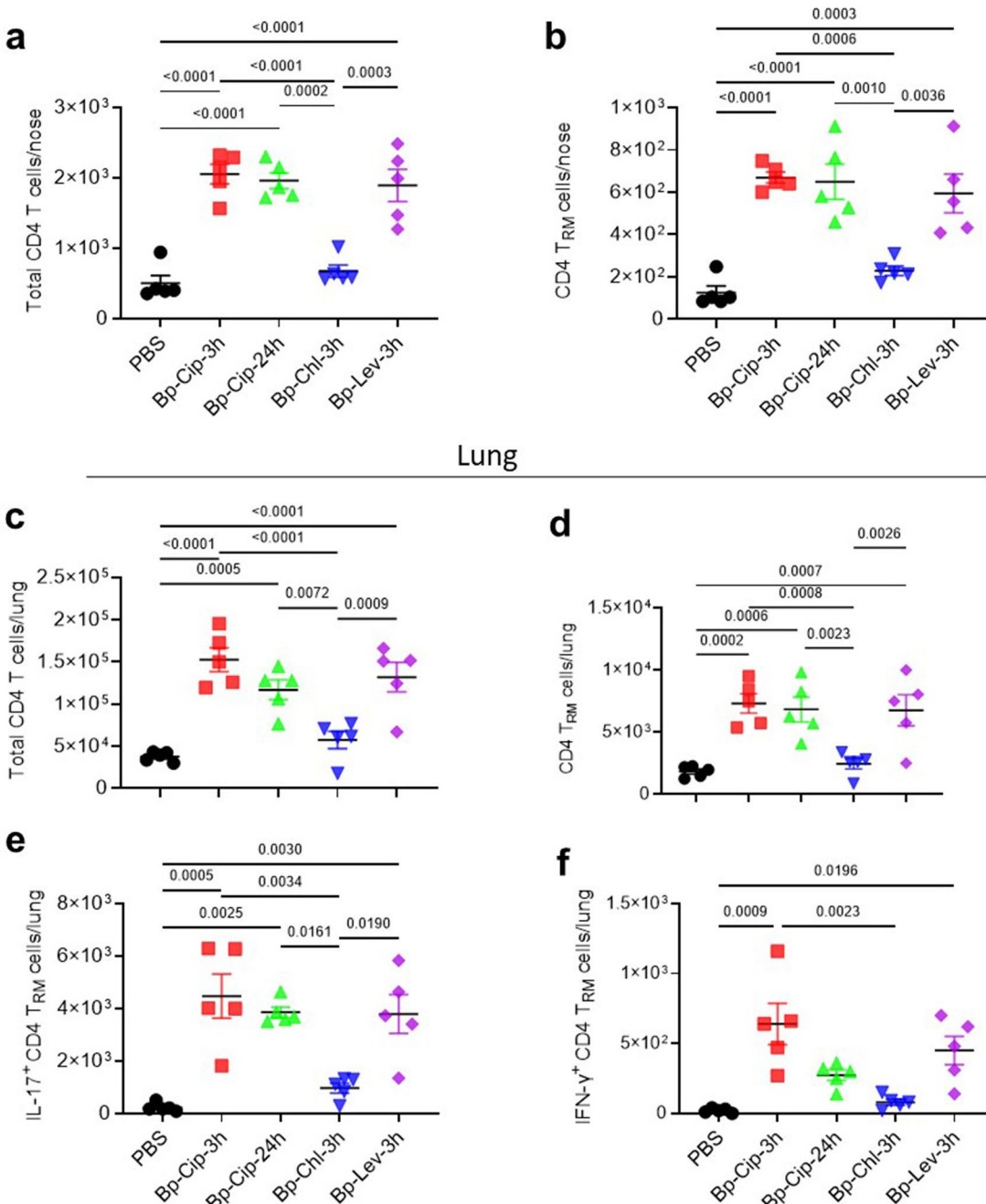

**Extended Data Fig. 4 | *B. pertussis* inactivated with ciprofloxacin or levofloxacin, but not chloramphenicol, induces CD4 T$_{RM}$ cells in nasal tissue and lungs of aerosol immunized mice.** *B. pertussis* were inactivated with 0.25 mg/ml ciprofloxacin for 3 h (Bp-Cip-3h) or 24 h (Bp-Cip-24h), 1 mg/ml levofloxacin for 3 h (Bp-Lev-3h) or 100 μg/mL chloramphenicol for 3 h (Bp-Chl-3h). Female 6-8 week old C57BL/6 mice were immunized once by aerosol administration of the antibiotic inactivated *B. pertussis* preparations or with PBS. 14 days post immunization, mice were injected i.v. with anti-CD45 antibody 10 min prior to euthanasia (to identify tissue-resident cells), and lung or nasal tissue cells were stained with antibodies specific for T$_{RM}$ cells. Mean number of CD4 T cells (**a**) or CD4 T$_{RM}$ cells (CD45iv⁻ CD4⁺CD44⁺CD62L⁻CD69⁺CD103$^{+/-}$) (**b**) in nasal tissue and mean number of CD4 T cells (**c**) or CD4 T$_{RM}$ cells (**d**) in lung by flow cytometric analysis. Data are mean ± s.e.m. (*n* = 5). Lung cells were stimulated for 16 hours with HKBP (10⁵ CFU/mL), anti-CD28 and anti-CD49d (1 μg/mL), followed by Brefeldin A (5 μg/ml) for the final 4 h of culture. *B. pertussis*-specific IL-17-producing (**e**) and IFNγ-producing (**f**) CD4 T$_{RM}$ cells were analyzed by ICS and flow cytometric analysis. Data are presented as mean ± SEM for biological replicates shown as individual symbols (*n* = 5). Data were analysed by Two-way ANOVA followed by Tukey's test for multiple comparisons. *P* values are shown above relevant data sets.

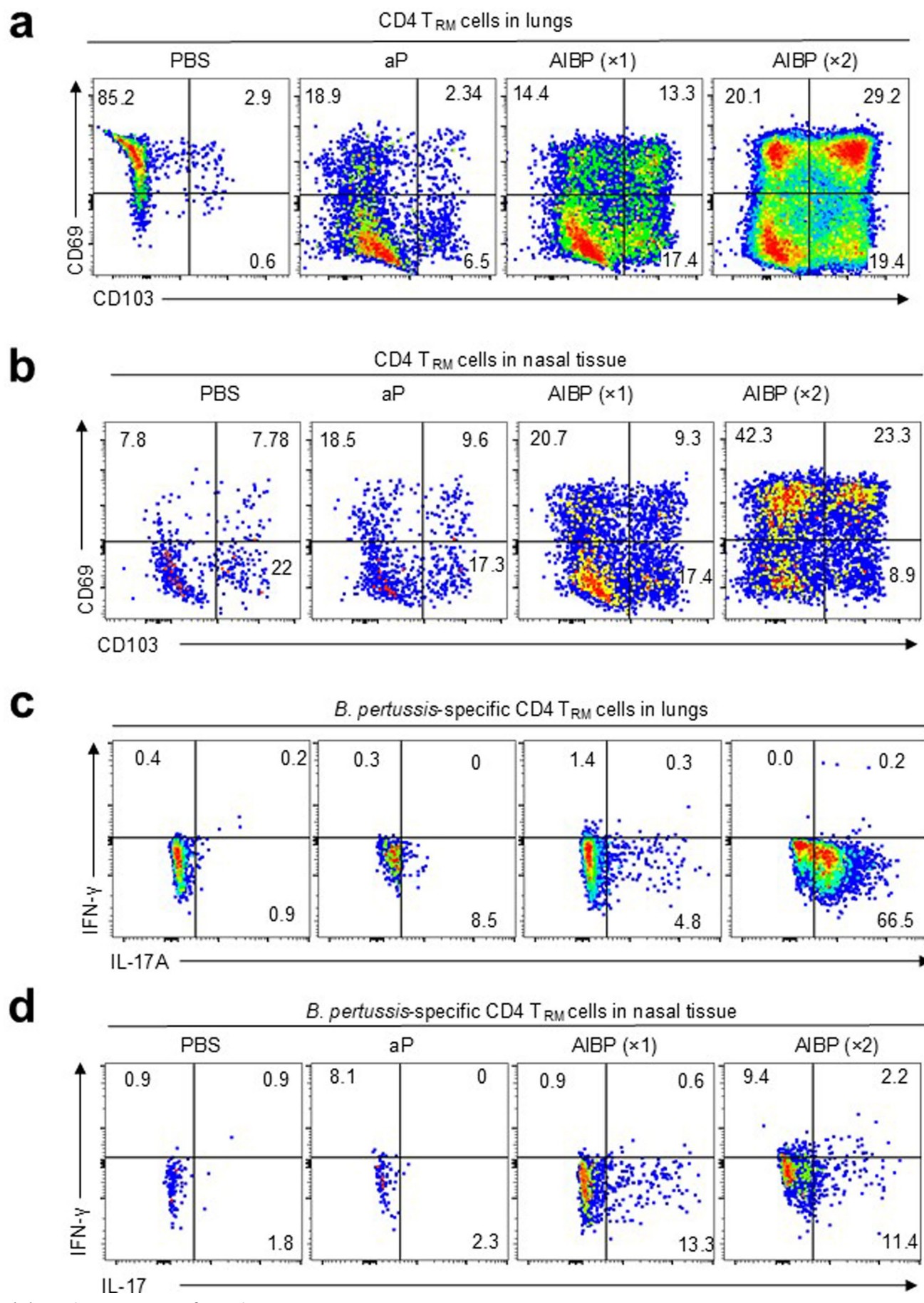

**Extended Data Fig. 5 | See next page for caption.**

**Extended Data Fig. 5 | Aerosol-delivered AIBP vaccine induces *B. pertussis*-specific $T_H1$- and $T_H17$-type $T_{RM}$ cells in the lungs and nasal tissue.** Female 6-8 week old C57BL/6 mice were immunized by aerosol administration of AIBP vaccine (once or twice at 0 and 4 weeks), i.m. immunization with an aP vaccine (twice, 0 and 4 weeks; 1/50 of the human dose) or PBS. Mice were aerosol challenged from a culture at $1 \times 10^9$ CFU/mL of live *B. pertussis* at week 6. On the day of but prior to challenge with live *B. pertussis*, mice were injected i.v. with anti-CD45 antibody 10 min prior to euthanasia (to identify tissue-resident cells), and lung or nasal tissue cells were stained with antibodies specific for $T_{RM}$ cells, or cells were stimulated with HKBP, anti-CD28 and anti-CD49d (both 1 μg/mL) for 16 h, followed by Brefeldin A (5 μg/ml) for the final 4 h of culture prior to intracellular cytokine staining (ICS) and flow cytometric analysis. Representative flow plots of CD69 and CD103 expression on CD45iv⁻CD4⁺CD44⁺CD62L⁻ T cells in lung tissue (**a**) or nasal tissue (**b**). Representative flow plots of *B. pertussis*-specific IFNγ- or IL-17-secreting CD45iv⁻CD4⁺ $T_{RM}$ cells in lung tissue (**c**) or nasal tissue (**d**).

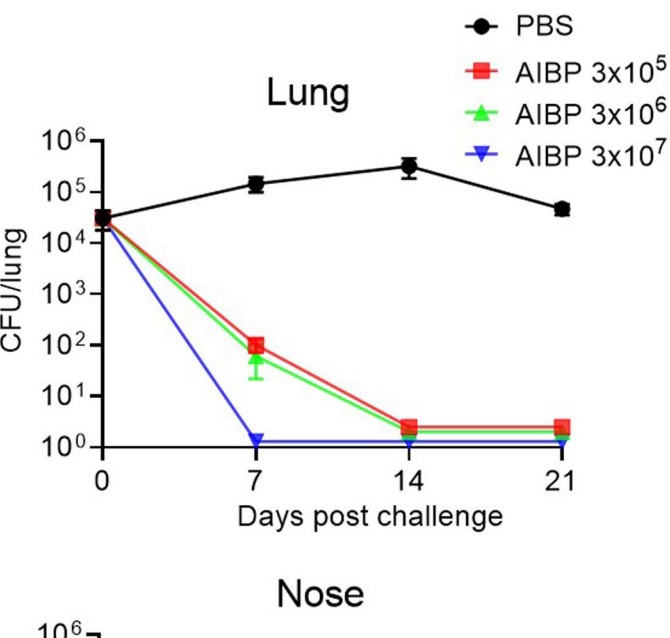

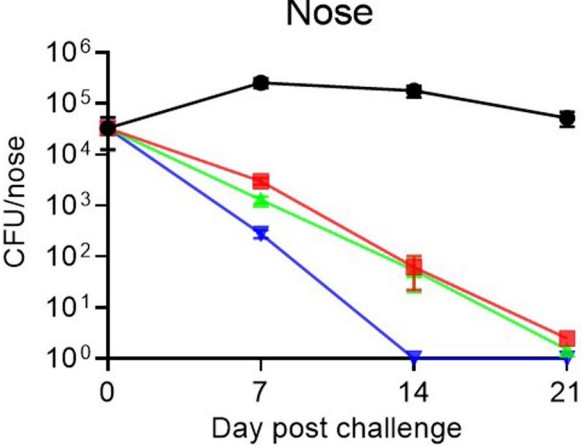

**Extended Data Fig. 6 | Intranasal administration of AIBP vaccine confers protection against infection with *B. pertussis* of the lung and nose.** Female 6-8 week old C57BL/6 mice were immunized i.n. with doses equivalent to $3\times10^5$, $3\times10^6$ or $3\times10^7$ CFU of AIBP vaccine (twice at 0 and 4 weeks) or PBS and challenged at week 6 by exposure to an aerosol from a culture containing $1\times10^9$ CFU/mL live *B. pertussis*. Live bacterial loads in lung and nasal tissue quantified by CFU counts 7, 14 and 21 days post live *B. pertussis* challenge. Data are presented as mean ± s.e.m. for biological replicates (*n* = 5). *P* values are shown above relevant data sets.

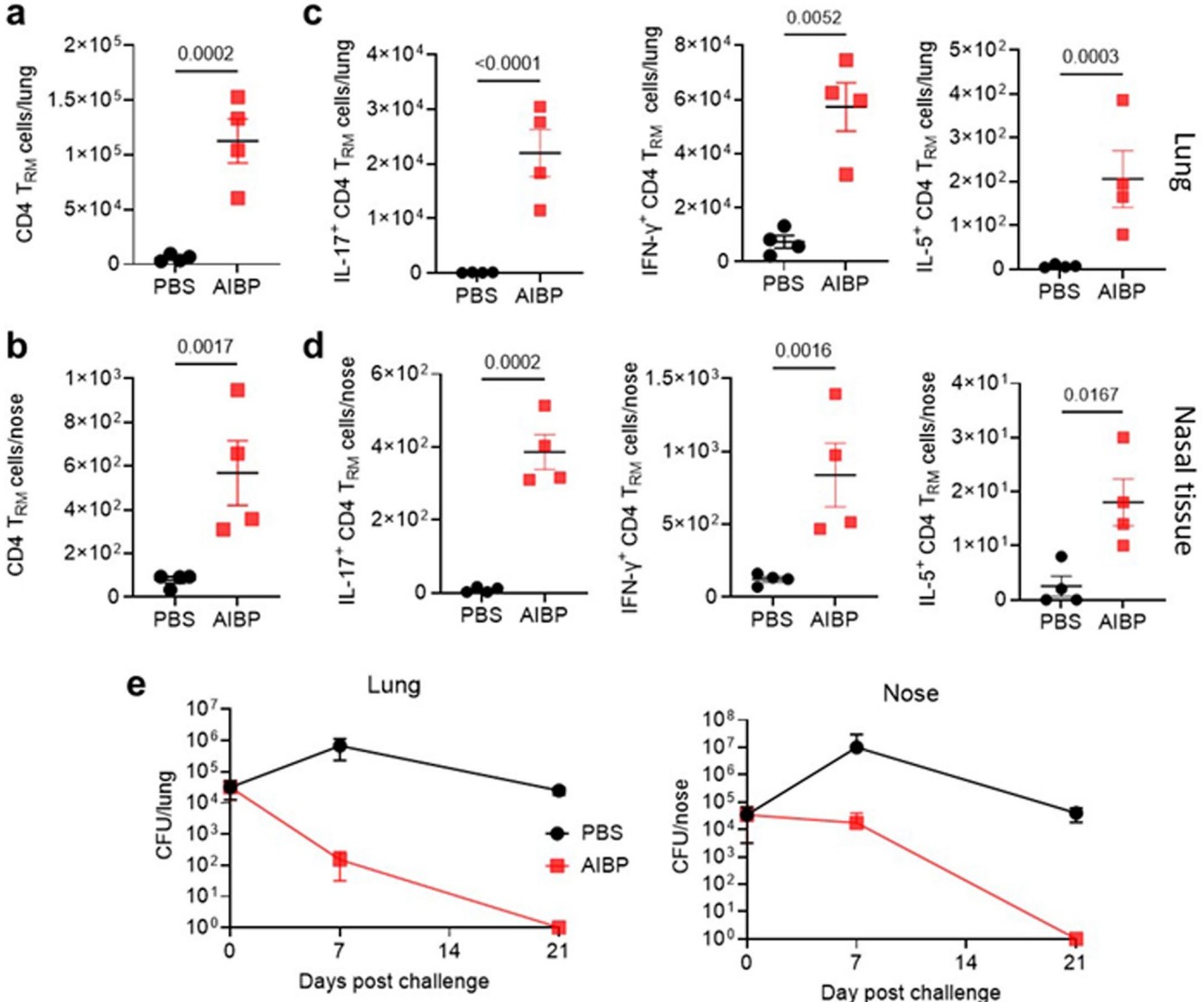

**Extended Data Fig. 7 | The AIBP vaccine induces persistent T$_{RM}$ cells in the respiratory tissue and long-term protective immunity against lung and nasal infection with *B. pertussis.*** Female 6-8 week old C57BL/6 mice were immunized by aerosol administration of AIBP vaccine (twice at 0 and 4 weeks) or PBS. Mice were aerosol challenged from a culture at 1×10$^9$ CFU/mL of live *B. pertussis* 6 months later. Seven days post *B. pertussis* challenge, mice were injected i.v. with anti-CD45 antibody 10 min prior to euthanasia (to identify tissue-resident cells), and lung or nasal tissue cells were stained with antibodies specific for T$_{RM}$ cells, or cells were stimulated with HKBP, anti-CD28 and anti-CD49d (both 1 µg/mL) for 16 h, followed by Brefeldin A (5 µg/ml) for the final 4 h of culture prior to ICS and

flow cytometric analysis. Mean absolute number of CD4 T$_{RM}$ cells (CD45iv⁻ CD4⁺ CD44⁺CD62L⁻CD69⁺CD103⁺/⁻ quantified in lungs (**a**) and nasal tissue (**b**) by flow cytometric analysis. Data are mean ± s.e.m. (*n* = 4). Number of *B. pertussis*-specific IFN-γ-, IL-17 or IL-5-secreting CD45iv⁻CD4 T cells in lungs (**c**) and nasal tissues (**d**) by ICS and flow cytometry. Data in **a-d** are presented as mean ± s.e.m. for biological replicates shown as individual symbols (*n* = 4). **e**, Live bacterial loads in lung and nose quantified by CFU counts 7 and 21 days post live *B. pertussis* challenge. Data are presented as mean ± s.e.m. for biological replicates (*n* = 4). Data were analyzed by one-way ANOVA followed by Tukey's test for multiple comparisons. *P* values are shown above relevant data sets.

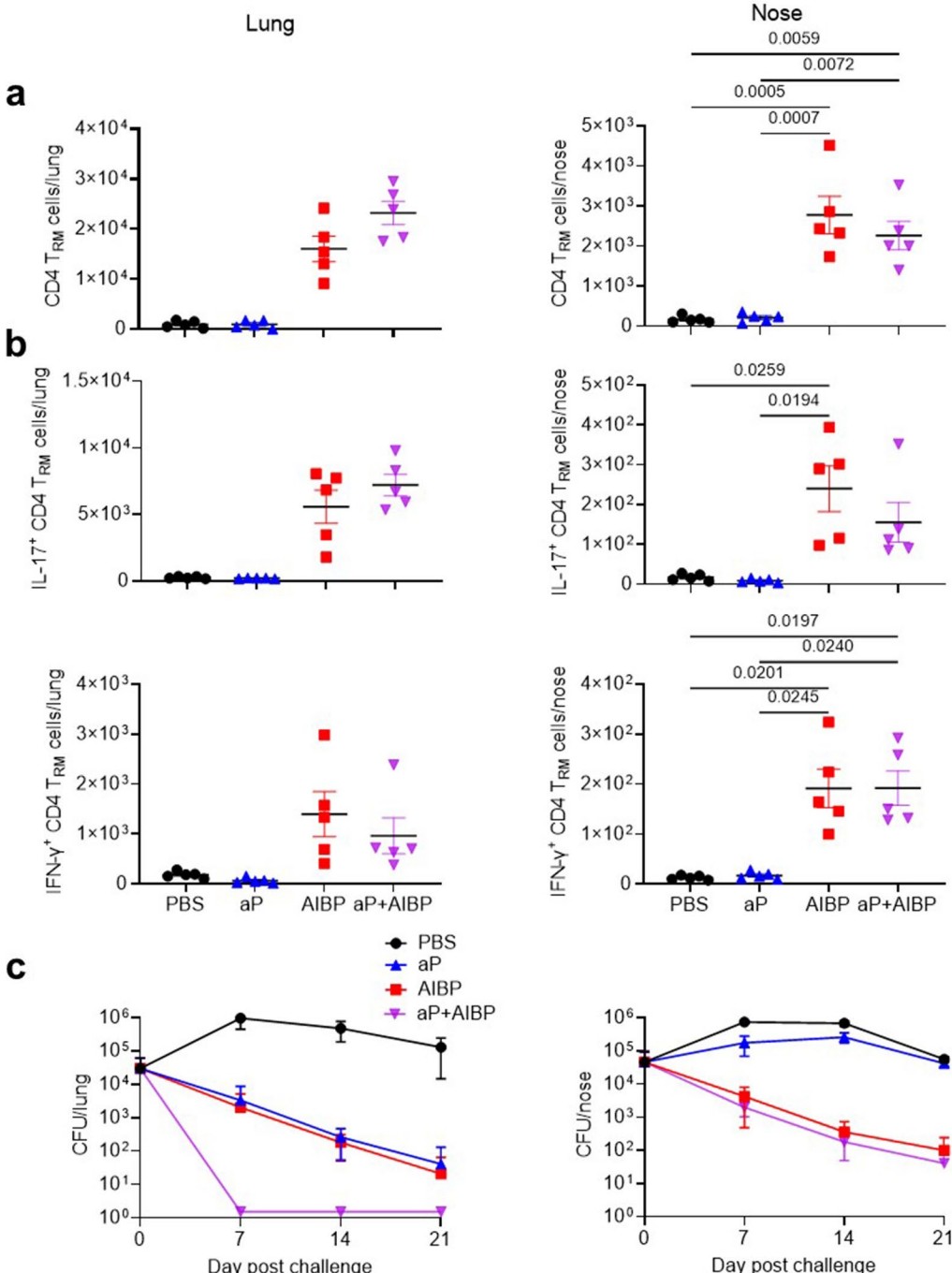

**Extended Data Fig. 8 | A single dose of the AIBP vaccine promotes induction of IL-17- and IFNγ-producing CD4 T$_{RM}$ cell and confers protective immunity against infection in mice previously immunized with two doses of an aP vaccine.** Female 6-8 week old C57BL/6 mice were immunized i.m. with aP vaccine (twice at 0 and 4 weeks; 1/50 of the human dose) followed by a single immunization of AIBP vaccine by aerosol at 8 weeks) or given the aP or AIBP only or PBS. Mice were aerosol challenged with live *B. pertussis* at week 14. **a**, On the day of but prior to challenge with live *B. pertussis*, CD4 T$_{RM}$ cells

(CD45iv⁻CD4⁺CD44⁺CD62L⁻CD69⁺CD103⁺/⁻) were quantified in lungs and nasal tissue by flow cytometry analysis; data are mean ± s.e.m. (*n* = 5). **b**, *B. pertussis*-specific IL-17- and IFNγ-producing CD4 T$_{RM}$ cell were analyzed by ICS and flow cytometry. Data in **a**,**b** are presented as mean ± s.e.m. for biological replicates shown as individual symbols (*n* = 5). **c**, CFU counts on lung and nasal tissue 7, 14 and 21 days post challenge. Data are presented as mean ± s.d. (*n* = 5). Data were analyzed by two-way ANOVAs followed by Tukey's test for multiple comparisons. *P* values are shown above relevant data sets.

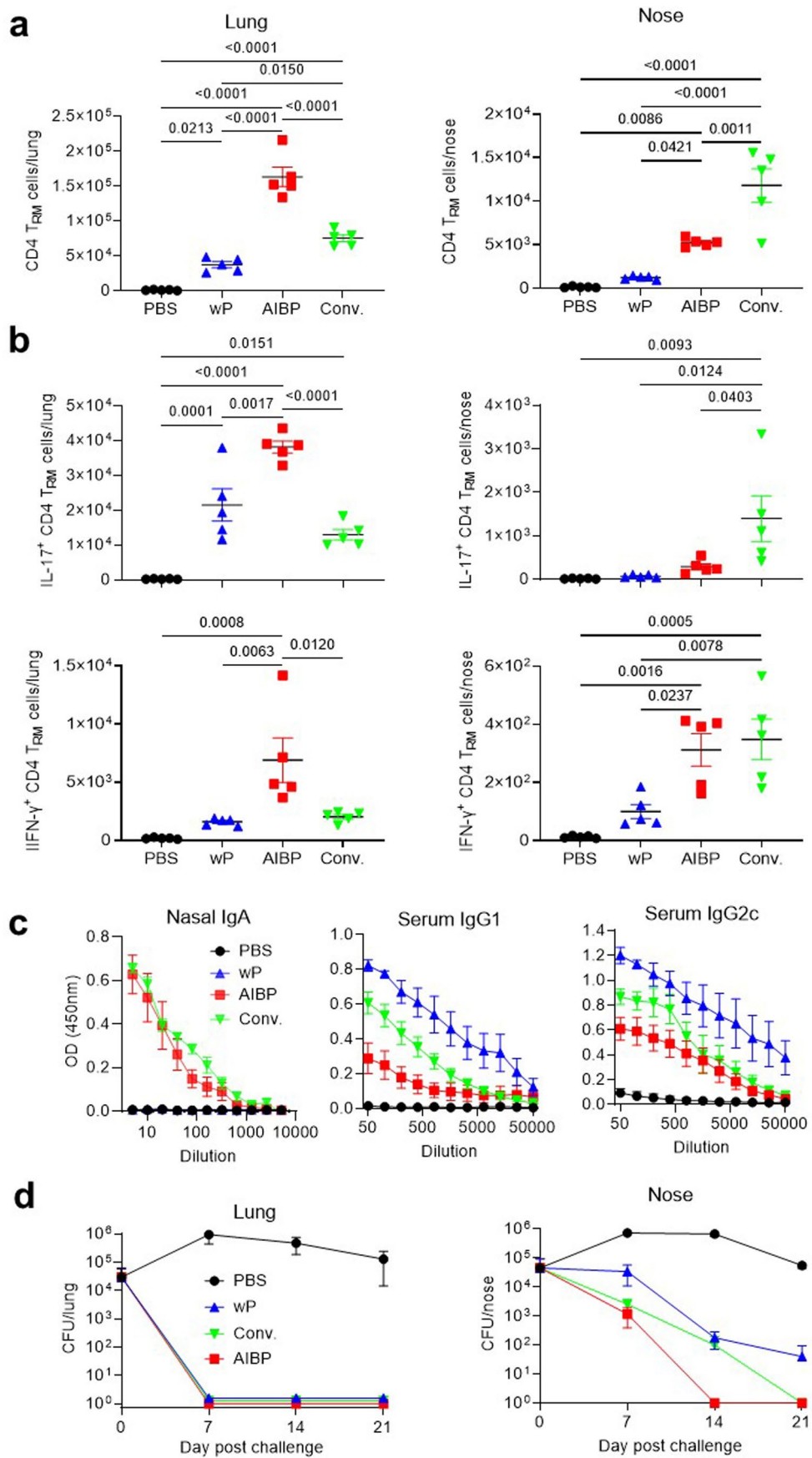

**Extended Data Fig. 9 | See next page for caption.**

**Extended Data Fig. 9 | The AIBP vaccine confers a higher level of protection against nasal infection than a parenterally-delivered wP vaccine or previous infection.** Female 6-8 week old C57BL/6 mice were immunized by aerosol administration of AIBP vaccine or i.m. administration of a wP vaccine (1/50 of the human dose) or PBS at 0 and 4 weeks or were infected with virulent *B. pertussis* and allowed to clear the infection. Mice were aerosol challenged with live *B. pertussis* at week 6. **a**, On the day of challenge, CD4 $T_{RM}$ cells (CD45iv⁻ CD4⁺CD44⁺CD62L⁻CD69⁺CD103⁺/⁻ were quantified in lungs and nasal tissue by flow cytometric analysis. **b**, Number of *B. pertussis*-specific IFNγ- or IL-17-secreting CD45iv⁻CD4 $T_{RM}$ cells in lungs and nasal tissues by ICS and flow cytometry. Data in **a** and **b** are presented as mean ± s.e.m. for biological replicates shown as individual symbols (*n* = 5). **c**, On the day prior to *B. pertussis* challenge, concentrations of *B. pertussis*-specific IgA in nasal tissue homogenates and *B. pertussis*-specific IgG1 and IgG2c in serum were quantified by ELISA. **d**, CFU counts on lung and nasal tissue 7, 14 and 21 days post challenge. Data in **c** and **d** are presented as mean ± s.e.m. for biological replicates (*n* = 5). The data for the groups of mice immunized with AIBP only or PBS are duplicated from Fig. 4. Data were analyzed by one-way ANOVA followed by Tukey's test for multiple comparisons. *P* values are shown above relevant datasets.

# Reporting Summary

## Statistics

For all statistical analyses, confirm that the following items are present in the figure legend, table legend, main text, or Methods section.

| n/a | Confirmed | |
|---|---|---|
| ☐ | ☒ | The exact sample size (*n*) for each experimental group/condition, given as a discrete number and unit of measurement |
| ☐ | ☒ | A statement on whether measurements were taken from distinct samples or whether the same sample was measured repeatedly |
| ☐ | ☒ | The statistical test(s) used AND whether they are one- or two-sided<br>*Only common tests should be described solely by name; describe more complex techniques in the Methods section.* |
| ☐ | ☒ | A description of all covariates tested |
| ☐ | ☒ | A description of any assumptions or corrections, such as tests of normality and adjustment for multiple comparisons |
| ☐ | ☒ | A full description of the statistical parameters including central tendency (e.g. means) or other basic estimates (e.g. regression coefficient) AND variation (e.g. standard deviation) or associated estimates of uncertainty (e.g. confidence intervals) |
| ☐ | ☒ | For null hypothesis testing, the test statistic (e.g. *F*, *t*, *r*) with confidence intervals, effect sizes, degrees of freedom and *P* value noted<br>*Give P values as exact values whenever suitable.* |
| ☒ | ☐ | For Bayesian analysis, information on the choice of priors and Markov chain Monte Carlo settings |
| ☒ | ☐ | For hierarchical and complex designs, identification of the appropriate level for tests and full reporting of outcomes |
| ☒ | ☐ | Estimates of effect sizes (e.g. Cohen's *d*, Pearson's *r*), indicating how they were calculated |

*Our web collection on statistics for biologists contains articles on many of the points above.*

## Software and code

Policy information about availability of computer code

| Data collection | Not applicable |
|---|---|
| Data analysis | Statistical analyses were performed using Graph-Pad Prism 9.0 Software. |

For manuscripts utilizing custom algorithms or software that are central to the research but not yet described in published literature, software must be made available to editors and reviewers. We strongly encourage code deposition in a community repository (e.g. GitHub). See the Nature Portfolio guidelines for submitting code & software for further information.

## Data

Policy information about availability of data

All manuscripts must include a data availability statement. This statement should provide the following information, where applicable:
- Accession codes, unique identifiers, or web links for publicly available datasets
- A description of any restrictions on data availability
- For clinical datasets or third party data, please ensure that the statement adheres to our policy

All data are available on request.

# Research involving human participants, their data, or biological material

Policy information about studies with human participants or human data. See also policy information about sex, gender (identity/presentation), and sexual orientation and race, ethnicity and racism.

| | |
|---|---|
| Reporting on sex and gender | Not applicable |
| Reporting on race, ethnicity, or other socially relevant groupings | Not applicable |
| Population characteristics | Not applicable |
| Recruitment | Not applicable |
| Ethics oversight | Not applicable |

Note that full information on the approval of the study protocol must also be provided in the manuscript.

# Field-specific reporting

Please select the one below that is the best fit for your research. If you are not sure, read the appropriate sections before making your selection.

☒ Life sciences          ☐ Behavioural & social sciences          ☐ Ecological, evolutionary & environmental sciences

For a reference copy of the document with all sections, see nature.com/documents/nr-reporting-summary-flat.pdf

# Life sciences study design

All studies must disclose on these points even when the disclosure is negative.

| | |
|---|---|
| Sample size | Sample sizes were chosen based on findings from similar previous experiments. |
| Data exclusions | No data were excluded from the analysis. |
| Replication | Each experiments was carried out 2-3 times with consistent and reproducible results. We have no reason to believe that reproducibility is an issue. |
| Randomization | C57BL/6 mice ( 6-8 weeks old) were randomly allocated to experimental groups. |
| Blinding | Mice were randomly allocated to experimental groups. Experimenters were unaware of the origin of cells and sample during analysis although no formal arrangements were in place. |

# Reporting for specific materials, systems and methods

We require information from authors about some types of materials, experimental systems and methods used in many studies. Here, indicate whether each material, system or method listed is relevant to your study. If you are not sure if a list item applies to your research, read the appropriate section before selecting a response.

## Materials & experimental systems

| n/a | Involved in the study |
|---|---|
| ☐ | ☒ Antibodies |
| ☒ | ☐ Eukaryotic cell lines |
| ☒ | ☐ Palaeontology and archaeology |
| ☐ | ☒ Animals and other organisms |
| ☒ | ☐ Clinical data |
| ☒ | ☐ Dual use research of concern |
| ☒ | ☐ Plants |

## Methods

| n/a | Involved in the study |
|---|---|
| ☒ | ☐ ChIP-seq |
| ☐ | ☒ Flow cytometry |
| ☒ | ☐ MRI-based neuroimaging |

# Antibodies

| | |
|---|---|
| Antibodies used | The details of antibodies used are provided in the Methods section and in Supplementary Table 1. |

| Validation | Antibodies were chosen based on manufacturer's website and previous published studies from our lab. |

## Animals and other research organisms

Policy information about <u>studies involving animals</u>; <u>ARRIVE guidelines</u> recommended for reporting animal research, and <u>Sex and Gender in Research</u>

| Laboratory animals | Female mice of the C57BL/6 strain were obtained from Charles River U.K. Mice were 6-8 weeks old at the initiation of experiments and housed in a specific pathogen-free facility in the Comparative Medicine Unit (CMU), Trinity College Dublin. |
| Wild animals | Not applicable |
| Reporting on sex | Findings apply to Female mice only. |
| Field-collected samples | Not applicable |
| Ethics oversight | All animal experiments were conducted according to the guidelines and under licenses approved by the Health Products Regulatory Authority (HPRA) of Ireland in accordance with prior ethical approval from Trinity College Dublin Animal Research Ethics Committee. |

Note that full information on the approval of the study protocol must also be provided in the manuscript.

## Plants

| Seed stocks | Not applicable |
| Novel plant genotypes | Not applicable |
| Authentication | Not applicable |

## Flow Cytometry

### Plots

Confirm that:

☐ The axis labels state the marker and fluorochrome used (e.g. CD4-FITC).

☐ The axis scales are clearly visible. Include numbers along axes only for bottom left plot of group (a 'group' is an analysis of identical markers).

☒ All plots are contour plots with outliers or pseudocolor plots.

☒ A numerical value for number of cells or percentage (with statistics) is provided.

### Methodology

| Sample preparation | Lungs and nasal tissues were digested and mashed through 70 μm strainer, red blood cells were lysed with ammonium chloride-potassium (ACK) buffer, cells were centrifuged at 1350 rpm, and used for flow cytometric stainig. |
| Instrument | Flow cytometric analysis was performed on an Aurora. |
| Software | Data were acquired using SpectroFlo. Data were analysed using FlowJo software (Tree Star). |
| Cell population abundance | Not applicable |
| Gating strategy | SSC-H/FSC-H-FSC-H/FSC-A-SSC-S/Live dead-CD8/CD4-CD45iv/CD4+-CD44+/CD62L-CD69+/CD103+-/-IL17-FNg+. Details provided in Supplementary Fig. 3. |

☒ Tick this box to confirm that a figure exemplifying the gating strategy is provided in the Supplementary Information.

