## [Peer Review File · Nature Microbiology]

Respiratory-delivered antibiotic-treated bacterial vaccine confers T cell-mediated sterilizing immunity against *Bordetella pertussis* in a murine model

Corresponding Author: Professor Kingston Mills

Version 0:

Reviewer comments:

Reviewer #1

(Remarks to the Author)

This manuscript by Jazayeri et al described the development and characterization of a pertussis vaccine composed of ciprofloxacin-treated *B. pertussis*. The authors characterize the immune response and test protection using a mouse model. The experiments show that two doses of the vaccine induce a Th1/Th17-mediated immune response, and results in clearance of *B. pertussis* from the nasal cavity and lungs at 14 days post-challenge. Importantly, the data also show that the vaccine is efficacious in mice previously vaccinated with the parenterally-administered acellular vaccine. Overall, this is a very nice study. The experiments are rigorous and well-controlled. The conclusions are justified by the data and the manuscript is well-written. My only/main criticism is that the authors should make it clearer throughout the manuscript what is known to be true in humans versus what is inferred based on experiments conducted with mice – and they should qualify their conclusions to indicate that they are drawing conclusions based on mouse data and whether the same conclusions will be true in humans is unknown.

Specific criticisms:

1. The title should include “in a murine model”
2. Line 24: “complete protection” should be defined. The data show that *B. pertussis* is cleared within 14 days post-challenge.
3. Lines 46-49: This sentence is confusing. Aim was to provide proof-of-principle against pertussis? An unmet medical need for a new vaccine and well-established animal models?
4. Lines 67-71: It should be made clearer in this section what conclusions are drawn from human studies versus mouse or baboon studies.
5. Line 76: define “completely protected”
6. Line 160: a reference is needed for this statement
7. Lines 185-187: a word is missing, or perhaps an extra word has been include, in this sentence
8. Throughout the section titled “Efficacy of AIBP vaccine not affected by priming with an aP vaccine”, the authors should describe the data and their interpretation of those data, rather than just stating the conclusions.
9. Lines 225-229: please make it clear what is known about human infection/immunity versus what is inferred based on experiments conducted with mice
10. Line 250: what is meant by “latency phase”?

Reviewer #2

(Remarks to the Author)

This is a very interesting study. The authors described a new vaccine approach that uses respiratory delivery of antibiotic-inactivated *Bordetella pertussis* (AIBP) to induce strong T cell responses and protection against lung and nasal infections. This vaccine strategy activates antigen-presenting cells, leading to the generation of *B. pertussis*-specific IL-17-producing CD4 tissue-resident memory (TRM) cells that help recruit neutrophils to the respiratory tract. Immunization with AIBP provided complete protection, which was dependent on CD4 T cells and IL-17. In contrast to traditional whole cell pertussis vaccines, the AIBP vaccine did not induce systemic pro-inflammatory cytokines, suggesting it is a safe and effective platform for inducing T cell-mediated immunity against respiratory pathogens.

Major comments:

Although this study delivers new insights, I believe it fits much better with NPJ Vaccines than with Nature Microbiology, simply because the focus is on vaccine research and immunological responses. The microbiological work described in this manuscript is limited to very superficial microscopy analyses.

Despite the fact that the authors describe a new vaccine approach it remains enigmatic why antibiotic-inactivated *Bordetella pertussis* showed stronger mucosal responses and better protection than the whole cell pertussis vaccine, while they essentially contain the same components. In the beginning of the study the authors compared the different formulations (AIBP and wP)

using different routes of vaccination, respiratory route (aerosolized) and intramuscular injection (results depicted in Figure 3), which is justifiable but complicates the comparison. In the second part of the study the authors compared the two formulations using the same (respiratory) route, showing that AIBP works slightly better (Figure 4). In the same context it is also unclear why for the DC maturation study (Figure 2) heat killed *B. pertussis* was used. It would have been better for the consistency of the story if the authors would have used wP.

Reviewer #3

(Remarks to the Author)

Reviewer Comments

The study from Jazayeri et al. represents an important advancement in pertussis research, successfully presenting a proof-of-concept for a vaccine (AIBP) that provides sterilizing immunity and generates local mucosal immunity, with engagement of both arms of adaptive immunity (i.e., with IgA and TRM secreting IL-17 responses), without systemic pro-inflammatory responses. The results from this vaccination approach are unparalleled by any currently licensed vaccines or other experimental vaccination strategies in the last decade, both in pre-clinical and clinical settings. This group is known for their expertise and pivotal findings in mouse models that have helped elucidate the immune responses to *B. pertussis*. The strength of this work, besides the novelty mentioned above, resides in the examination of multiple components of immunity, including innate immunity, and the benchmarking with different vaccine platforms and administration schemes including priming, boosting, and challenge. This allows for extrapolation of observations to diverse scenarios that would likely be present at the population level in a human clinical setting. I believe the manuscript could be improved by addressing the suggestions below:

General Comments

- A doubt that remains in this study is whether current wP vaccines (or any inactivated whole-cell vaccine) could exert the exact same activity if the mechanism of delivery were the same as the AIBP vaccine, or if the use of the antibiotic to treat BP and the "latent" state achieved is the key for the superior immunity. I think this would be very important to clarify.
- Would a different type of antibiotic produce the same results?
- While some safety data is presented (no PT detection, limited inflammatory response), more comprehensive toxicology studies would strengthen the manuscript. The long-term safety of antibiotic-inactivated bacteria requires also further evaluation.
- From an organizational point of view, this study lacks coherence in the study design. For example, in the different experimental settings, there is no consistency in the vaccination scheme that is compared to the AIBP vaccination, jumping back and forth between aP vaccination, then wP vaccination, and then aP vaccination, etc., or in the immunological readouts. A more standardized comparison of immune parameters across all conditions would clarify relative advantages.

Specific Comments

Abstract

The reader gets the impression that only a comparison with wP vaccination was performed. The aP vaccination should also be mentioned.

Introduction

It would be interesting to develop the concept of aerosolized or intranasal delivery a bit more. Has this type of approach been used already in clinical settings? And if so, with what success?

Line 69: "However, while current parenterally delivered alum adjuvanted aP vaccines induce potent circulating serum IgG antibodies and Th2-biased responses, they do not induce IgA, Th1, Th17 or respiratory TRM cells" - This statement saying that aP vaccines do not induce Th1 or other of those components may be too extreme, especially for Th1 responses that have been consistently observed in human responses. I think mentioning "lower levels" instead of "do not induce" would be more accurate.

Results

- Overall, in the challenging experiments, in this reviewer's opinion, it would make more sense to report the results of protection from challenge after setting up the immunological readouts from vaccination that are measured immediately before the challenge (e.g., for Figure 3, panel a would come last).
- Figure 1B – The conclusion about the morphology according to the authors does not seem substantiated by the microscopy image shown. The treated bacteria sample seems to show aggregates rather than individual elongated bacteria. Better quality images with nuclear staining should be provided. Quantification, for example with a flow cytometry approach for size and complexity and a viability dye, would further validate the authors' claims.
- Figure 1D- Since the values reported in PBS are the "background" of the detection of PT, shouldn't the data be reported instead as the background-subtracted data?
- What is the relevance of studying IgG2c?

- Please define HKBP.
- Figure 3 – It would be interesting to know the breakdown of where the cytokines are predominantly coming from, i.e., are they mainly associated with CD69+CD103+, CD69+CD103-, CD69-CD103+, or both?
- Sup. Fig 4 – What is the explanation for the sharp decrease in % of CD69+ cells in vaccination compared to PBS?
- Line 173: "IL-17- or IFN- γ -secreting B. pertussis-specific CD4 TRM cells were significantly higher in the lungs and nasal tissue of mice immunized with the AIBP vaccine compared with the wP vaccine or previous infection (Supplementary Fig. 6b)." – This observation is only true for the lungs. In the nasal tissue, it is only higher compared to wP but not previous infection.
- Lines 185/186 – The authors mention that existing aP vaccines promote B. pertussis-specific Th2, but Th2 responses are never shown in this paper. More than rediscovering the wheel for aP vaccines, it would be interesting to know if the AIBP vaccine would also induce Th2 responses.
- Line 231 – "in mice immunized with one or two doses of the AIBP vaccine." The authors wrote that mice were immunized with one or two doses, but only the results of one dose are depicted.

Discussion

The study doesn't address how long protection lasts. Given that waning immunity is a significant issue with current aP vaccines, data on durability of protection would be valuable.

Overall, this study presents a promising new approach to pertussis vaccination with solid immunological data supporting its mechanism of action.

Decision Letter:

25th April 2025

Dear Professor Mills,

Thank you for your patience while your manuscript "Respiratory-delivered antibiotic-inactivated bacterial vaccine confers T cell-mediated sterilizing immunity against *Bordetella pertussis*" was under peer-review at Nature Microbiology. It has now been seen by 3 referees, whose expertise and comments you will find at the end of this email. Although they find your work of some potential interest, they have raised a number of concerns that will need to be addressed before we can consider publication of the work in Nature Microbiology.

Should further experimental data allow you to address the reviewers' criticisms, we would be happy to look at a revised manuscript.

Please include a data availability statement as a separate section after Methods but before references, under the heading "Data Availability". This section should inform readers about the availability of the data used to support the conclusions of your study. This information includes accession codes to public repositories (data banks for protein, DNA or RNA sequences, microarray, proteomics data etc...), references to source data published alongside the paper, unique identifiers such as URLs to data repository entries, or data set DOIs, and any other statement about data availability. At a minimum, you should include the following statement: "The data that support the findings of this study are available from the corresponding author upon request", mentioning any restrictions on availability. If DOIs are provided, we also strongly encourage including these in the Reference list (authors, title, publisher (repository name), identifier, year). For more guidance on how to write this section please see: <http://www.nature.com/authors/policies/data/data-availability-statements-data-citations.pdf>

* If you have not done so already we suggest that you begin to revise your manuscript so that it conforms to our Article format

instructions at <http://www.nature.com/nmicrobiol/info/final-submission>. Refer also to any guidelines provided in this letter.

When submitting the revised version of your manuscript, please pay close attention to our [href="https://www.nature.com/nature-portfolio/editorial-policies/image-integrity">Digital Image Integrity Guidelines.](https://www.nature.com/nature-portfolio/editorial-policies/image-integrity) and to the following points below:

EXTENDED DATA FIGURES

Link Redacted

Note: This url links to your confidential homepage and associated information about manuscripts you may have submitted or be reviewing for us. If you wish to forward this e-mail to co-authors, please delete this link to your homepage first.

Nature Microbiology is committed to improving transparency in authorship. As part of our efforts in this direction, we are now requesting that all authors identified as 'corresponding author' on published papers create and link their Open Researcher and Contributor Identifier (ORCID) with their account on the Manuscript Tracking System (MTS), prior to acceptance. This applies to primary research papers only. ORCID helps the scientific community achieve unambiguous attribution of all scholarly contributions. You can create and link your ORCID from the home page of the MTS by clicking on 'Modify my Springer Nature account'. For more information please visit www.springernature.com/orcid.

If you wish to submit a suitably revised manuscript we would hope to receive it within 6 months. If you cannot send it within this time, please let us know. We will be happy to consider your revision, even if a similar study has been accepted for publication at Nature Microbiology or published elsewhere (up to a maximum of 6 months).

Yours sincerely,

Reviewer Expertise:

- Referee #1: pertussis pathogenesis.
- Referee #2: nasal vaccines, bacterial vaccines.
- Referee #3: Pertussis and T cells and vaccines.

Reviewer Comments:

Reviewer #1 (Remarks to the Author):

This manuscript by Jazayeri et al described the development and characterization of a pertussis vaccine composed of ciprofloxacin-treated *B. pertussis*. The authors characterize the immune response and test protection using a mouse model. The experiments show that two doses of the vaccine induce a Th1/Th17-mediated immune response, and results in clearance of *B. pertussis* from the nasal cavity and lungs at 14 days post-challenge. Importantly, the data also show that the vaccine is efficacious in mice previously vaccinated with the parenterally-administered acellular vaccine. Overall, this is a very nice study. The experiments are rigorous and well-controlled. The conclusions are justified by the data and the manuscript is well-written. My only/main criticism is that the authors should make it clearer throughout the manuscript what is known to be true in humans versus what is inferred based on experiments conducted with mice – and they should qualify their conclusions to indicate that they are drawing conclusions based on mouse data and whether the same conclusions will be true in humans is unknown.

Specific criticisms:

1. The title should include "in a murine model"

2. Line 24: "complete protection" should be defined. The data show that B. pertussis is cleared within 14 days post-challenge.
3. Lines 46-49: This sentence is confusing. Aim was to provide proof-of-principle against pertussis? An unmet medical need for a new vaccine and well-established animal models?
4. Lines 67-71: It should be made clearer in this section what conclusions are drawn from human studies versus mouse or baboon studies.
5. Line 76: define "completely protected"
6. Line 160: a reference is needed for this statement
7. Lines 185-187: a word is missing, or perhaps an extra word has been include, in this sentence
8. Throughout the section titled "Efficacy of AIBP vaccine not affected by priming with an aP vaccine", the authors should describe the data and their interpretation of those data, rather than just stating the conclusions.
9. Lines 225-229: please make it clear what is known about human infection/immunity versus what is inferred based on experiments conducted with mice
10. Line 250: what is meant by "latency phase"?

Reviewer #2 (Remarks to the Author):

This is a very interesting study. The authors described a new vaccine approach that uses respiratory delivery of antibiotic-inactivated *Bordetella pertussis* (AIBP) to induce strong T cell responses and protection against lung and nasal infections. This vaccine strategy activates antigen-presenting cells, leading to the generation of B. pertussis-specific IL-17-producing CD4 tissue-resident memory (TRM) cells that help recruit neutrophils to the respiratory tract. Immunization with AIBP provided complete protection, which was dependent on CD4 T cells and IL-17. In contrast to traditional whole cell pertussis vaccines, the AIBP vaccine did not induce systemic pro-inflammatory cytokines, suggesting it is a safe and effective platform for inducing T cell-mediated immunity against respiratory pathogens.

Major comments:

Although this study delivers new insights, I believe it fits much better with NPJ Vaccines than with Nature Microbiology, simply because the focus is on vaccine research and immunological responses. The microbiological work described in this manuscript is limited to very superficial microscopy analyses.

Despite the fact that the authors describe a new vaccine approach it remains enigmatic why antibiotic-inactivated *Bordetella pertussis* showed stronger mucosal responses and better protection than the whole cell pertussis vaccine, while they essentially contain the same components. In the beginning of the study the authors compared the different formulations (AIBP and wP) using different routes of vaccination, respiratory route (aerosolized) and intramuscular injection (results depicted in Figure 3), which is justifiable but complicates the comparison. In the second part of the study the authors compared the two formulations using the same (respiratory) route, showing that AIBP works slightly better (Figure 4). In the same context it is also unclear why for the DC maturation study (Figure 2) heat killed B. pertussis was used. It would have been better for the consistency of the story if the authors would have used wP.

Reviewer #3 (Remarks to the Author):

Reviewer Comments

The study from Jazayeri et al. represents an important advancement in pertussis research, successfully presenting a proof-of-concept for a vaccine (AIBP) that provides sterilizing immunity and generates local mucosal immunity, with engagement of both arms of adaptive immunity (i.e., with IgA and TRM secreting IL-17 responses), without systemic pro-inflammatory responses. The results from this vaccination approach are unparalleled by any currently licensed vaccines or other experimental vaccination strategies in the last decade, both in pre-clinical and clinical settings. This group is known for their expertise and pivotal findings in mouse models that have helped elucidate the immune responses to B. pertussis. The strength of this work, besides the novelty mentioned above, resides in the examination of multiple components of immunity, including innate immunity, and the benchmarking with different vaccine platforms and administration schemes including priming, boosting, and challenge. This allows for extrapolation of observations to diverse scenarios that would likely be present at the population level in a human clinical setting. I believe the manuscript could be improved by addressing the suggestions below:

General Comments

- A doubt that remains in this study is whether current wP vaccines (or any inactivated whole-cell vaccine) could exert the exact same activity if the mechanism of delivery were the same as the AIBP vaccine, or if the use of the antibiotic to treat BP and the "latent" state achieved is the key for the superior immunity. I think this would be very important to clarify.
- Would a different type of antibiotic produce the same results?
- While some safety data is presented (no PT detection, limited inflammatory response), more comprehensive toxicology studies would strengthen the manuscript. The long-term safety of antibiotic-inactivated bacteria requires also further evaluation.
- From an organizational point of view, this study lacks coherence in the study design. For example, in the different experimental settings, there is no consistency in the vaccination scheme that is compared to the AIBP vaccination, jumping back and forth between aP vaccination, then wP vaccination, and then aP vaccination, etc., or in the immunological readouts. A more standardized comparison of immune parameters across all conditions would clarify relative advantages.

Specific Comments

Abstract

The reader gets the impression that only a comparison with wP vaccination was performed. The aP vaccination should also be mentioned.

Introduction

It would be interesting to develop the concept of aerosolized or intranasal delivery a bit more. Has this type of approach been used already in clinical settings? And if so, with what success?

Line 69: "However, while current parenterally delivered alum adjuvanted aP vaccines induce potent circulating serum IgG antibodies and Th2-biased responses, they do not induce IgA, Th1, Th17 or respiratory TRM cells" - This statement saying that aP vaccines do not induce Th1 or other of those components may be too extreme, especially for Th1 responses that have been consistently observed in human responses. I think mentioning "lower levels" instead of "do not induce" would be more accurate.

Results

- Overall, in the challenging experiments, in this reviewer's opinion, it would make more sense to report the results of protection from challenge after setting up the immunological readouts from vaccination that are measured immediately before the challenge (e.g., for Figure 3, panel a would come last).

- Figure 1B – The conclusion about the morphology according to the authors does not seem substantiated by the microscopy image shown. The treated bacteria sample seems to show aggregates rather than individual elongated bacteria. Better quality images with nuclear staining should be provided. Quantification, for example with a flow cytometry approach for size and complexity and a viability dye, would further validate the authors' claims.

- Figure 1D- Since the values reported in PBS are the "background" of the detection of PT, shouldn't the data be reported instead as the background-subtracted data?

- What is the relevance of studying IgG2c?

- Please define HKBP.

- Figure 3 – It would be interesting to know the breakdown of where the cytokines are predominantly coming from, i.e., are they mainly associated with CD69+CD103+, CD69+CD103-, CD69-CD103+, or both?

- Sup. Fig 4 – What is the explanation for the sharp decrease in % of CD69+ cells in vaccination compared to PBS?

- Line 173: "IL-17- or IFN- γ -secreting B. pertussis-specific CD4 TRM cells were significantly higher in the lungs and nasal tissue of mice immunized with the AIBP vaccine compared with the wP vaccine or previous infection (Supplementary Fig. 6b)." – This observation is only true for the lungs. In the nasal tissue, it is only higher compared to wP but not previous infection.

- Lines 185/186 – The authors mention that existing aP vaccines promote B. pertussis-specific Th2, but Th2 responses are never shown in this paper. More than rediscovering the wheel for aP vaccines, it would be interesting to know if the AIBP vaccine would also induce Th2 responses.

- Line 231 – "in mice immunized with one or two doses of the AIBP vaccine." The authors wrote that mice were immunized with one or two doses, but only the results of one dose are depicted.

Discussion

The study doesn't address how long protection lasts. Given that waning immunity is a significant issue with current aP vaccines, data on durability of protection would be valuable.

Overall, this study presents a promising new approach to pertussis vaccination with solid immunological data supporting its mechanism of action.

Version 1:

Reviewer comments:

Reviewer #1

(Remarks to the Author)

In this revised manuscript, Jazayeri et al have addressed the concerns raised by the reviewers adequately. I have only a few minor critiques:

1. Throughout the manuscript, sentences begin with "In contrast,". I believe that used this way, the term should be "By contrast," ("In contrast" is used with "In contrast to. . .")
2. On lines 52-53, the sentence seems to indicate that there is an unmet need for well-established animal models.
3. Line 59: the word "vaccine" should be added after (aP).
4. Line 93: delete "with"?
5. Lines 330-333: should include "at least in mice"

Reviewer #2

(Remarks to the Author)

Dear editor,

I am satisfied with the responses provided by the authors.

I agree that formaldehyde treatment is the most likely explanation of the differences observed between wP and AIBP vaccines, resulting in the superior performance of the AIBP vaccine. Specifically, AIBP appears more effective than wP in promoting dendritic cell maturation and the production of Th1- and Th17-polarizing cytokines.

One minor nuance I would like to highlight is that the duration of formaldehyde treatment significantly influences the extent of protein antigen cross-linking. This effect may vary among different wP formulations and could contribute to variability in immunogenicity.

Best regards,

Marien

Reviewer #3

(Remarks to the Author)

I would like to commend the authors for thoroughly addressing all my comments as well as those of the other reviewers.

I believe the manuscript has been greatly improved and have no further questions or concerns.

This work represents, in my opinion, a very important milestone in pertussis research.

Decision Letter:

Our ref: NMICROBIOL-25010347A

20th August 2025

Dear Dr. Mills,

Thank you for submitting your revised manuscript "Respiratory-delivered antibiotic-treated bacterial vaccine confers T cell-mediated sterilizing immunity against *Bordetella pertussis* in a murine model" (NMICROBIOL-25010347A). It has now been seen by the original referees and their comments are below. The reviewers find that the paper has improved in revision, and therefore we'll be happy in principle to publish it in Nature Microbiology, pending minor revisions to satisfy the referees' final requests and to comply with our editorial and formatting guidelines.

Thank you again for your interest in Nature Microbiology Please do not hesitate to contact me if you have any questions.

Sincerely,

Reviewer #1 (Remarks to the Author):

In this revised manuscript, Jazayeri et al have addressed the concerns raised by the reviewers adequately. I have only a few minor critiques:

1. Throughout the manuscript, sentences begin with "In contrast,". I believe that used this way, the term should be "By contrast," ("In contrast" is used with "In contrast to. . .")
2. On lines 52-53, the sentence seems to indicate that there is an unmet need for well-established animal models.
3. Line 59: the word "vaccine" should be added after (aP).
4. Line 93: delete "with"?

5.Lines 330-333: should include "at least in mice"

Reviewer #2 (Remarks to the Author):

I am satisfied with the responses provided by the authors.

I agree that formaldehyde treatment is the most likely explanation of the differences observed between wP and AIBP vaccines, resulting in the superior performance of the AIBP vaccine. Specifically, AIBP appears more effective than wP in promoting dendritic cell maturation and the production of Th1- and Th17-polarizing cytokines.

One minor nuance I would like to highlight is that the duration of formaldehyde treatment significantly influences the extent of protein antigen cross-linking. This effect may vary among different wP formulations and could contribute to variability in immunogenicity.

Reviewer #3 (Remarks to the Author):

I would like to commend the authors for thoroughly addressing all my comments as well as those of the other reviewers.

I believe the manuscript has been greatly improved and have no further questions or concerns.

This work represents, in my opinion, a very important milestone in pertussis research.

Version 2:

Decision Letter:

30th September 2025

Dear Professor Mills,

I am pleased to accept your Article "Respiratory-delivered antibiotic-treated bacterial vaccine confers T cell-mediated sterilizing immunity against Bordetella pertussis in a murine model" for publication in Nature Microbiology. Thank you for having chosen to submit your work to us and many congratulations.

You may wish to make your media relations office aware of your accepted publication, in case they consider it appropriate to organize some internal or external publicity. Once your paper has been scheduled you will receive an email confirming the publication details. This is normally 3-4 working days in advance of publication. If you need additional notice of the date and time of publication, please let the production team know when you receive the proof of your article to ensure there is sufficient time to coordinate. Further information on our embargo policies can be found here:

<https://www.nature.com/authors/policies/embargo.html>

Authors may need to take specific actions to achieve compliance with funder and institutional open access mandates. If your research is supported by a funder that requires immediate open access (e.g. according to [a href="https://www.springernature.com/gp/open-science/plan-s-compliance"> Plan S principles](https://www.springernature.com/gp/open-science/plan-s-compliance) or the [a href="https://www.springernature.com/gp/open-science/us-federal-agency-compliance"> NIH public access policy](https://www.springernature.com/gp/open-science/us-federal-agency-compliance)) then you should select the gold OA route, and we will direct you to the compliant route where possible. Because authors warrant under our subscription licensing terms that they haven't committed to licensing any version of their article under a licence inconsistent with the terms of our agreement – including the applicable embargo period – publication under the subscription model isn't suitable for authors whose funders require no embargo.

With kind regards,

P.S. Click on the following link if you would like to recommend Nature Microbiology to your librarian
<http://www.nature.com/subscriptions/recommend.html#forms>

** Visit the Springer Nature Editorial and Publishing website at http://editorial-jobs.springernature.com?utm_source=ejP_NMicro_email&utm_medium=ejP_NMicro_email&utm_campaign=ejP_NMicro for more information about our career opportunities. If you have any questions please click [here](mailto:editorial.publishing.jobs@springernature.com).

Open Access This Peer Review File is licensed under a Creative Commons Attribution 4.0 International License, which permits use, sharing, adaptation, distribution and reproduction in any medium or format, as long as you give appropriate credit to the original author(s) and the source, provide a link to the Creative Commons license, and indicate if changes were made. In cases where reviewers are anonymous, credit should be given to 'Anonymous Referee' and the source. The images or other third party material in this Peer Review File are included in the article's Creative Commons license, unless indicated otherwise in a credit line to the material. If material is not included in the article's Creative Commons license and your intended use is not permitted by statutory regulation or exceeds the permitted use, you will need to obtain permission directly

from the copyright holder.

Response to Referees

Reviewer #1 (Remarks to the Author):

This manuscript by Jazayeri et al described the development and characterization of a pertussis vaccine composed of ciprofloxacin-treated B. pertussis. The authors characterize the immune response and test protection using a mouse model. The experiments show that two doses of the vaccine induce a Th1/Th17-mediated immune response, and results in clearance of B. pertussis from the nasal cavity and lungs at 14 days post-challenge. Importantly, the data also show that the vaccine is efficacious in mice previously vaccinated with the parenterally-administered acellular vaccine. Overall, this is a very nice study. The experiments are rigorous and well-controlled. The conclusions are justified by the data and the manuscript is well-written. My only/main criticism is that the authors should make it clearer throughout the manuscript what is known to be true in humans versus what is inferred based on experiments conducted with mice – and they should qualify their conclusions to indicate that they are drawing conclusions based on mouse data and whether the same conclusions will be true in humans is unknown.

Author response: We thank the reviewer for the positive comments on our manuscript.

Specific criticisms:

1. *The title should include “in a murine model”*

Author response: We have added “in a murine model” to the title.

2. *Line 24: “complete protection” should be defined. The data show that B. pertussis is cleared within 14 days post-challenge.*

Author response: We have amended this statement to read a “high level of protection”.

3. *Lines 46-49: This sentence is confusing. Aim was to provide proof-of-principle against pertussis? An unmet medical need for a new vaccine and well-established animal models?*

Author response: We have amended this sentence.

4. *Lines 67-71: It should be made clearer in this section what conclusions are drawn from human studies versus mouse or baboon studies.*

Author response: We have amended this section to distinguish the findings from mouse, baboon and human studies

5. *Line 76: define “completely protected”*

Author response: We have amended this from “completely protected” to “protected”.

6. *Line 160: a reference is needed for this statement*

Author response: We have added references and amended the sentence slightly to reflect these references.

7. *Lines 185-187: a word is missing, or perhaps an extra word has been include, in this sentence*

Author response: We have edited and split this sentence to clarify the points we wished to make.

8. *Throughout the section titled “Efficacy of AIBP vaccine not affected by priming with an aP*

vaccine”, the authors should describe the data and their interpretation of those data, rather than just stating the conclusions.

Author response: We have edited this section to describe the data rather than stating the conclusions.

9. Lines 225-229: please make it clear what is known about human infection/immunity versus what is inferred based on experiments conducted with mice

Author response: We have clarified which comments refer to mouse, baboon or human studies.

10. Line 250: what is meant by “latency phase”?

Author response: We were referring to the period soon after antibiotic treatment, but on reflection we think that the term “latency phase” is not a widely used term in this context and have removed it. Instead have stated the times after antibiotic treatment. In response to comment from reviewer #3, we have examined the immunogenicity of the AIBP vaccine prepared by ciprofloxacin-treatment of *B. pertussis* for 3 hours, compared with 24 hours. This data has been added to Extended Data Fig. 4.

Reviewer #2 (Remarks to the Author):

This is a very interesting study. The authors described a new vaccine approach that uses respiratory delivery of antibiotic-inactivated Bordetella pertussis (AIBP) to induce strong T cell responses and protection against lung and nasal infections. This vaccine strategy activates antigen-presenting cells, leading to the generation of B. pertussis-specific IL-17-producing CD4 tissue-resident memory (TRM) cells that help recruit neutrophils to the respiratory tract. Immunization with AIBP provided complete protection, which was dependent on CD4 T cells and IL-17. In contrast to traditional whole cell pertussis vaccines, the AIBP vaccine did not induce systemic pro-inflammatory cytokines, suggesting it is a safe and effective platform for inducing T cell-mediated immunity against respiratory pathogens.

Author response: We thank the reviewer for the positive comments on our manuscript.
Major comments:

Although this study delivers new insights, I believe it fits much better with NPJ Vaccines than with Nature Microbiology, simply because the focus is on vaccine research and immunological responses. The microbiological work described in this manuscript is limited to very superficial microscopy analyses.

Author response: “Vaccines” are listed in the aims and scope of the journal as one of the topics for papers for consideration by *Nature Microbiology*. Indeed, the journal has published several hundred articles on vaccines.

Despite the fact that the authors describe a new vaccine approach it remains enigmatic why antibiotic-inactivated Bordetella pertussis showed stronger mucosal responses and better protection than the whole cell pertussis vaccine, while they essentially contain the same components. In the beginning of the study the authors compared the different formulations (AIBP and wP) using different routes of vaccination, respiratory route (aerosolized) and intramuscular injection (results depicted in Figure 3), which is justifiable but complicates the comparison. In the second part of the study the authors compared the two formulations using the same (respiratory) route, showing that AIBP works slightly better (Figure 4). In the same context it is also unclear why for the DC maturation study (Figure 2) heat killed B. pertussis

was used. It would have been better for the consistency of the story if the authors would have used wP.

Author response: The AIBP vaccine has very distinct characteristics and properties compared with the wP vaccines. wP vaccines are prepared by inactivating *B. pertussis* with aldehydes, usually formaldehyde, which cross links proteins by forming covalent bonds between different amino acid residues. This can adversely affect conformational antibody epitopes and alter protein degradation by endo-lysosomal proteases, thereby affecting antigen processing and T cell activation. This can affect the immunogenicity and protective efficacy of the killed bacteria in the wP vaccines. In contrast, the bacterial antigens are not modified in the AIBP vaccine and our data demonstrate that the AIBP vaccine is more effective than wP vaccine at stimulating APC and thereby promoting Th1 and Th17 responses in vitro. Furthermore, the AIBP vaccine induces significantly greater accumulation of Th1 and Th17-type TRM cells in the respiratory tissue and confers better protection against *B. pertussis* infection of the nasal cavity.

As suggested by the reviewer, we have now directly compared the effects of the AIBP vaccine with the wP vaccine on DC maturation, the production of Th1 and Th17 polarizing cytokines and activation APC to induce Th17 responses in vitro (Fig. 2 of revised manuscript). We have also directly compared the immunogenicity and protective efficacy the AIBP vaccine with the wP vaccine, where both vaccines are delivered by the intranasal route (Fig. 5 of revised manuscript). The data clearly demonstrate the superior immunogenicity and efficacy of the AIBP vaccine over the wP vaccine when both are delivered by i.n. route. This is consistent with the new data showing that the AIBP vaccine is more effective than the wP vaccine in inducing DC maturation and production of Th1 and Th17 polarizing cytokines. In addition to the new data directly comparing AIBP and wP vaccines, we have added comments on the merits of the AIBP over the wP vaccines to the discussion section of the revised manuscript.

Reviewer #3 (Remarks to the Author):

Reviewer Comments

*The study from Jazayeri et al. represents an important advancement in pertussis research, successfully presenting a proof-of-concept for a vaccine (AIBP) that provides sterilizing immunity and generates local mucosal immunity, with engagement of both arms of adaptive immunity (i.e., with IgA and TRM secreting IL-17 responses), without systemic pro-inflammatory responses. The results from this vaccination approach are unparalleled by any currently licensed vaccines or other experimental vaccination strategies in the last decade, both in pre-clinical and clinical settings. This group is known for their expertise and pivotal findings in mouse models that have helped elucidate the immune responses to *B. pertussis*. The strength of this work, besides the novelty mentioned above, resides in the examination of multiple components of immunity, including innate immunity, and the benchmarking with different vaccine platforms and administration schemes including priming, boosting, and challenge. This allows for extrapolation of observations to diverse scenarios that would likely be present at the population level in a human clinical setting. I believe the manuscript could be improved by addressing the suggestions below:*

Author response: We thank the reviewer for the very positive comments on our manuscript.

General Comments

- A doubt that remains in this study is whether current wP vaccines (or any inactivated whole-cell vaccine) could exert the exact same activity if the mechanism of delivery were the*

same as the AIBP vaccine, or if the use of the antibiotic to treat BP and the "latent" state achieved is the key for the superior immunity. I think this would be very important to clarify.

Author response: As suggested by the reviewer, we have now directly compared the wP and AIBP vaccines for immunogenicity and protective efficacy when both vaccines are delivered to mice by intranasal route (Fig. 5 in the revised manuscript). In response to a comment from reviewer #2, we have also compared the effects of wP and AIBP vaccines on DC maturation and production of Th1 and Th17 polarizing cytokines (Fig. 2 in the revised manuscript). The data clearly demonstrate the superior immunogenicity and efficacy of the AIBP vaccine over the wP vaccine when both are delivered by i.n. route. This is consistent with the new data showing that the AIBP vaccine is more effective than the wP vaccine in inducing DC maturation and production of Th1 and Th17 polarizing cytokines. In addition to the new data directly comparing AIBP and wP vaccines, we have added comments on the merits of the AIBP over the wP vaccines to the discussion section of the revised manuscript.

Regarding the question whether the latent state is key for superior immunity (in response to a comment from Reviewer #1, we have removed the term "latency phase"), we have compared the immune responses induced with bacteria 3 hours and 24 hours after antibiotic treatment and found that both preparations were immunogenic, with a marginal, but not significantly stronger T_{RM} responses with the bacteria treated for 3 hours. The new data shown in Extended Data Fig. 4 in the revised manuscript.

• *Would a different type of antibiotic produce the same results?*

Author response: Many antibiotics kill bacteria by inhibiting protein synthesis or by disrupting cell walls, which leads to lysis of the bacterial cells. We choose ciprofloxacin because it arrests cell division by inhibiting the function of DNA gyrase and/or DNA Topoisomerase IV leading to DNA damage. The bacteria are not lysed by the ciprofloxacin treatment, but are enlarged and this may facilitate uptake of the inactivated bacteria by APC. We have now examined *B. pertussis* inactivated with ciprofloxacin or levofloxacin (another fluoroquinolone family antibiotics) compared with chloramphenicol, a Beta-lactam antibiotic that inhibits protein synthesis. At concentrations that inhibited bacterial growth, ciprofloxacin or levofloxacin did not lyse the bacteria but caused some of the bacteria to elongate. In contrast, chloramphenicol-treated *B. pertussis* had lost their structural integrity, which was consistent with lysis of the bacteria (new data added to Extended Data Fig. 4). Aerosol immunization of mice with *B. pertussis* inactivated with ciprofloxacin for 3 or 24 hours or levofloxacin for 3 hours resulted in expansion of CD4 T_{RM} cells in the lungs and nasal tissue of immunized mice and *B. pertussis*-specific IFN- γ and IL-17-producing T cells in the lung and spleen. In contrast, the CD4 T cell responses in mice immunized with chloramphenicol-inactivated *B. pertussis* was at background levels, similar to that observed in mice immunized with PBS. (new data added to Extended Data Fig. 4). These findings suggest that *B. pertussis* inactivated with fluoroquinolone antibiotics, but not Beta-lactam antibiotics, are highly immunogenic *in vivo*, inducing *B. pertussis*-specific CD4 T cells in the respiratory tract when delivered by the aerosol route to mice.

• *While some safety data is presented (no PT detection, limited inflammatory response), more comprehensive toxicology studies would strengthen the manuscript. The long-term safety of antibiotic-inactivated bacteria requires also further evaluation.*

Author response: We have extended the safety testing to assessment of serum CRP concentrations, which are considered a marker of vaccine safety. We have also assessed circulating neutrophil and lymphocyte counts and body weight changes post vaccination. This data has been added to Fig 2 and Extended Data Fig. 2 of the revised manuscript.

• *From an organizational point of view, this study lacks coherence in the study design. For example, in the different experimental settings, there is no consistency in the vaccination scheme that is compared to the AIBP vaccination, jumping back and forth between aP vaccination, then wP vaccination, and then aP vaccination, etc., or in the immunological readouts. A more standardized comparison of immune parameters across all conditions would clarify relative advantages.*

Author response: We have reorganized the presentation of the data to make it more consistent between different experimental settings. We have also changed the order of figures 4 and 5, so that we show data that on a comparison of the AIBP vaccine with the aP vaccine in Fig 3, the effect of priming with the aP vaccine on response to AIBP vaccine in Fig. 4 and a comparison between AIBP and wP, when both are delivered by the i.n. route, in Fig. 5. The original Fig 5 showing a comparison of the AIBP vaccine delivered by aerosol with the wP vaccine delivered i.m. has been moved to Extended Data Fig 9 to make way for the new data in Fig 5.

Specific Comments

Abstract

The reader gets the impression that only a comparison with wP vaccination was performed. The aP vaccination should also be mentioned.

Author response: We have now mentioned the aP vaccine in the abstract.

Introduction

It would be interesting to develop the concept of aerosolized or intranasal delivery a bit more. Has this type of approach been used already in clinical settings? And if so, with what success?

Author response: We have added some discussion on this point.

Line 69: "However, while current parenterally delivered alum adjuvanted aP vaccines induce potent circulating serum IgG antibodies and Th2-biased responses, they do not induce IgA, Th1, Th17 or respiratory TRM cells" - This statement saying that aP vaccines do not induce Th1 or other of those components may be too extreme, especially for Th1 responses that have been consistently observed in human responses. I think mentioning "lower levels" instead of "do not induce" would be more accurate.

Author response: We agree re the induction of Th1 responses with aP vaccines and have edited this section

Results

• *Overall, in the challenging experiments, in this reviewer's opinion, it would make more sense to report the results of protection from challenge after setting up the immunological readouts from vaccination that are measured immediately before the challenge (e.g., for Figure 3, panel a would come last).*

Author response: We have re-organized the figures as suggested by the reviewer.

• *Figure 1B – The conclusion about the morphology according to the authors does not seem substantiated by the microscopy image shown. The treated bacteria sample seems to show*

aggregates rather than individual elongated bacteria. Better quality images with nuclear staining should be provided. Quantification, for example with a flow cytometry approach for size and complexity and a viability dye, would further validate the authors' claims.

Author response: We have now provided better quality images with nuclear staining of *B. pertussis* treated with ciprofloxacin or levofloxacin (another fluoroquinolone family antibiotic) compared with chloramphenicol, a Beta-lactam antibiotic. The morphology data has been moved to Extended Data Fig. 1; this allowed us to show larger images and to directly compare the effect of the three antibiotics with live untreated bacteria. The ciprofloxacin-treated bacteria had intact morphology and a proportion were elongated, but not lysed. Treatment of *B. pertussis* with levofloxacin had a similar effect on bacterial morphology. In contrast, *B. pertussis* treated with chloramphenicol had lost their structural integrity, consistent with lysis of the bacteria (Extended Data Fig. 1)

• Figure 1D- Since the values reported in PBS are the "background" of the detection of PT, shouldn't the data be reported instead as the background-subtracted data?

Author response: We believe that it is important to show the background values as this allows a statistical comparison between AIBP or wP vaccine and control as well as between with AIBP and wP vaccines.

• *What is the relevance of studying IgG2c?*

Author response: IgG2c and IgG2a antibodies play a role in complement fixation and opsonization, which facilitates phagocytosis of bacteria. IgG2a and IgG2c are driven by Th1 cells. C57BL/6 mice express IgG2c but not IgG2a and IgG2c antibodies are considered to have a protective role against *B. pertussis* in C57BL/6 mice. We have added comments to the text of the revised manuscript.

• *Please define HKBP.*

Author response: HKBP is heat-killed *Bordetella pertussis*. This is stated on page 5 of the revised manuscript.

• *Figure 3 – It would be interesting to know the breakdown of where the cytokines are predominantly coming from, i.e., are they mainly associated with CD69⁺CD103⁺, CD69⁺CD103⁻, CD69⁻CD103⁺, or both?*

Author response: We use CD69 or CD103 to determine the frequency of total CD4 T_{RM} cells in lung or nasal tissue. However, since CD69 expression can be enhanced following antigen-specific activation of T cells in vitro, CD69 or CD103 expression is not used to determine the frequency of antigen-specific cytokine-secreting CD4 T_{RM} cells. Instead, we gate on CD4 and CD45i.v. to quantify tissue-resident CD4 T cells and since only memory T cells produced cytokines in response to re-stimulation with antigen in vitro, when we then gate on IL-17⁺ or IFN-γ⁺ cells, this gives us a measure of cytokine-secreting *B. pertussis*-specific CD4 T_{RM} cells.

• *Sup. Fig 4 – What is the explanation for the sharp decrease in % of CD69⁺ cells in vaccination compared to PBS?*

Author response: The decrease in the frequency of CD69⁺CD103⁻ cells in the lungs of vaccinated mice is partly a reflection of the significant increase in the frequency of CD69⁺CD103⁺ cells, especially in mice given the AIBP vaccine (Extended Data Fig. 7a in revised manuscript). Furthermore, when expressed as absolute number, the total numbers of CD69⁺CD103^{+/-} cells is significantly increased in mice immunized with the AIBP vaccine (Fig. 3a in revised manuscript). In nasal tissue, the frequency of CD69⁺CD103⁻ are

dramatically increased in mice immunized with the AIBP vaccine (Extended Data Fig. 5b in revised manuscript).

• *Line 173: "IL-17- or IFN- γ -secreting *B. pertussis*-specific CD4 TRM cells were significantly higher in the lungs and nasal tissue of mice immunized with the AIBP vaccine compared with the wP vaccine or previous infection (Extended Data Fig. 6b)." – This observation is only true for the lungs. In the nasal tissue, it is only higher compared to wP but not previous infection.*

Author response: We have amended the text of the results in line with the reviewers' comments.

• *Lines 185/186 – The authors mention that existing aP vaccines promote *B. pertussis*-specific Th2, but Th2 responses are never shown in this paper. More than rediscovering the wheel for aP vaccines, it would be interesting to know if the AIBP vaccine would also induce Th2 responses.*

Author response: We have examined *B. pertussis*-specific IL-5 production in mice immunized with the AIBP and found that it does induce weak Th2 responses (results of long-term experiment shown in Extended Data Fig. 7), but induces stronger Th1 and Th17 responses, which are now considered to be central for protection against *B. pertussis*.

• *Line 231 – "in mice immunized with one or two doses of the AIBP vaccine." The authors wrote that mice were immunized with one or two doses, but only the results of one dose are depicted.*

Author response: We have corrected this error.

Discussion

The study doesn't address how long protection lasts. Given that waning immunity is a significant issue with current aP vaccines, data on durability of protection would be valuable.

Author response: We have assessed immunity induced with the AIBP vaccine 6 months after immunization (we had two group of mice that had been immunized with the AIBP vaccine or PBS approximately 3 months before receipt of the review of our manuscript). *B. pertussis* challenge of mice 6 months post immunization demonstrated a high level of protection against infection of the lungs and nose, with complete bacterial clearance by day 21 (Extended Data Fig. 7e,f). Furthermore, CD4 TRM cells were still elevated in lung and nasal tissues of mice 6 months after immunization with the AIBP vaccine (Extended Data Fig. 7a,b). In addition, substantial numbers *B. pertussis*-specific IL-17- and IFN- γ -producing CD4 TRM cells were detectable in lung. The new data demonstrates that the AIBP vaccine confers durable immunity against *B. pertussis*.

Overall, this study presents a promising new approach to pertussis vaccination with solid immunological data supporting its mechanism of action.

Author response: We thank the reviewer for their complementary comments on our study.

Responses to reviewers' comment

Reviewer #1:

Remarks to the Author:

In this revised manuscript, Jazayeri et al have addressed the concerns raised by the reviewers adequately. I have only a few minor critiques:

1. Throughout the manuscript, sentences begin with "In contrast,". I believe that used this way, the term should be "By contrast," ("In contrast" is used with "In contrast to. . .

Author response: We have changed "In contrast" to "By contract" throughout.

2. On lines 52-53, the sentence seems to indicate that there is an unmet need for well-established animal models.

Author response: We have amended this by splitting sentence into two.

3. Line 59: the word "vaccine" should be added after (aP).

Author response: We have added "vaccine" after aP.

4. Line 93: delete "with"?

Author response: We have deleted "with".

5. Lines 330-333: should include "at least in mice"

Author response: We have added "in mice" to the end of this sentence.

Reviewer #2:

Remarks to the Author:

I am satisfied with the responses provided by the authors.

I agree that formaldehyde treatment is the most likely explanation of the differences observed between wP and AIBP vaccines, resulting in the superior performance of the AIBP vaccine. Specifically, AIBP appears more effective than wP in promoting dendritic cell maturation and the production of Th1- and Th17-polarizing cytokines.

One minor nuance I would like to highlight is that the duration of formaldehyde treatment significantly influences the extent of protein antigen cross-linking. This effect may vary among different wP formulations and could contribute to variability in immunogenicity.

Author response: We have added a sentence in line with this comment from the reviewer.

Reviewer #3:

Remarks to the Author:

I would like to commend the authors for thoroughly addressing all my comments as well as those of the other reviewers.

I believe the manuscript has been greatly improved and have no further questions or concerns. This work represents, in my opinion, a very important milestone in pertussis research.

Author response: We are grateful to all the reviewers for their helpful and complementary comments.